# SIV-Bench: A Video Benchmark for Social Interaction Understanding and Reasoning

## Abstract

The rich and multifaceted nature of human social interaction, encompassing multimodal cues, unobservable relations and mental states, and dynamical behavior, presents a formidable challenge for artificial intelligence. To advance research in this area, we introduce SIV-Bench, a novel video benchmark for rigorously evaluating the capabilities of Multimodal Large Language Models (MLLMs) across Social Scene Understanding (SSU), Social State Reasoning (SSR), and Social Dynamics Prediction (SDP). SIV-Bench features 2,792 video clips and 8,792 meticulously generated question-answer pairs derived from a human-LLM collaborative pipeline. It is originally collected from TikTok and YouTube, covering a wide range of video genres, presentation styles, and linguistic and cultural backgrounds. It also includes a dedicated setup for analyzing the impact of different textual cues—original on-screen text, added dialogue, or no text. Our comprehensive experiments on leading MLLMs reveal that while models adeptly handle SSU, they significantly struggle with SSR and SDP, where Relation Inference (RI) is an acute bottleneck, as further examined in our analysis. Our study also confirms the critical role of transcribed dialogue, particularly in aiding the reasoning of social states and dynamics. By systematically identifying current MLLMs' strengths and limitations, SIV-Bench offers crucial insights to steer the development of more socially intelligent AI.

## 1 Introduction

The development of Multimodal Large Language Models (MLLMs), which can process text, images, and video, marks a significant step toward more human-like AI (Team et al., 2023; Wu et al., 2024; Zhu et al., 2025b; Hurst et al., 2024; Yang et al., 2024). These models perform strongly across a range of tasks, driving progress in areas such as visual reasoning, video captioning, and multimodal dialogue. As these capabilities expand, there is a growing need for benchmarks that can evaluate model performance, uncover limitations, and guide future research (Fu et al., 2024; Fang et al., 2024; Zhou et al., 2024; Qiang et al., 2025; Huang et al., 2024). One critical yet underexplored area is the understanding and reasoning about social interactions — a core aspect of social intelligence that encompasses not only observable behaviors but also implicit mental states and social relationships governing behaviors such as forming bonds, exchanging information, and coordinating actions (Berger et al., 1972; Smith-Lovin & Heise, 1988). However, current video benchmarks—whether designed for specific tasks like object segmentation (Hong et al., 2024; Xu et al., 2018), captioning (Qi et al., 2023), and activity recognition (Zhao et al., 2019), or for broader video understanding across various domains (Li et al., 2024b;c)—still lack a detailed focus on social interactions.

We define the capacity of MLLMs to understand and reason social interactions in video through three core, interrelated dimensions: **1) Social Scene Understanding (SSU)** is foundational, enabling the recognition of visible elements such as objects, environments, and socially salient human features like body movements, clothing, and physical appearance. Reliable scene perception is required to ground interpretations in relevant cues. **2) Social State Reasoning (SSR)** is essential for interpreting the unobservable states of interaction, such as emotions, intents, attitudes, and interpersonal relationships, which guide and shape behavior (Strachan et al., 2024; Wu et al., 2020). This capacity allows models to move beyond surface-level features and grasp the underlying states. **3) Social Dynamics Prediction (SDP)** enables the model to reason about how interactions proceed over time or under alternative conditions, capabilities essential for a flexible and human-like understanding of social scenarios (Ramnani & Miall, 2004; Byrne, 2016). It involves factual prediction and counterfactual

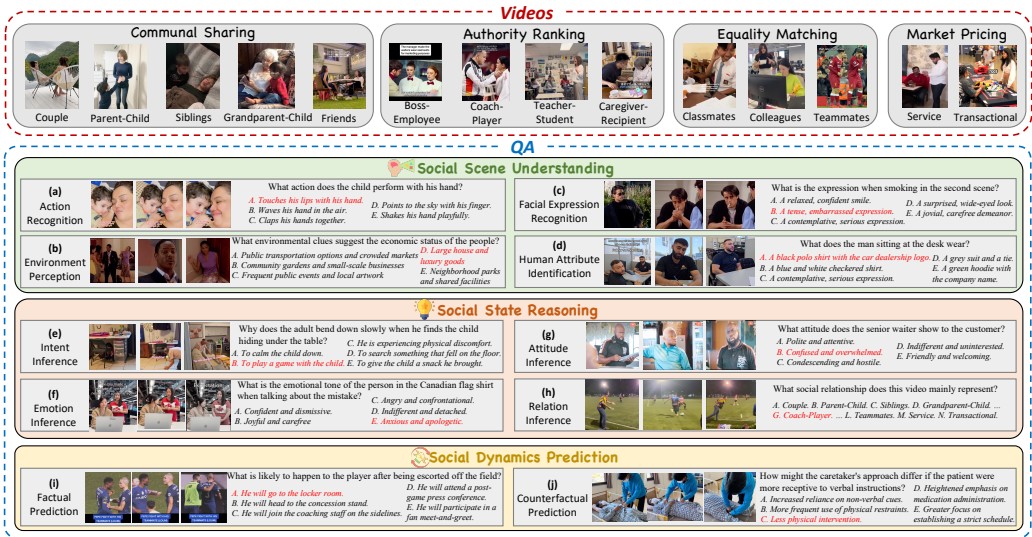

Figure 1: Overview of SIV-Bench, showing its diverse videos spanning various social interactions and sample QAs for three task dimensions: Social Scene Understanding (SSU), Social State Reasoning (SSR), and Social Dynamics Prediction (SDP), along with their fine-grained sub-tasks.

prediction. The former anticipates upcoming actions or emotional shifts. The latter examines how changes in social relations or behaviors might alter the outcome of an interaction.

In this work, we introduce **SIV-Bench** (**S**ocial **I**nteraction **V**ideo **Bench**mark), a novel benchmark designed to evaluate MLLMs across the above three core dimensions. SIV-Bench is fundamentally organized around social relationships, recognizing their critical role in shaping social interaction (Thibaut, 2017; Burkitt, 1997; Hartup, 1989). Specifically, SIV-Bench is built on Fiske's Relational Models Theory (Fiske, 1992), categorizing social interactions via four foundational models (*Communal Sharing*, *Authority Ranking*, *Equality Matching*, and *Market Pricing*), which are instantiated through 14 specific relation types (e.g., parent-child, friends, colleagues). This relational schema underpins fine-grained evaluation across three key dimensions. Understanding relational context is vital for SSU, as the salience and interpretation of actions, expressions, and environmental cues often depend on the relationship (e.g., a gaze between colleagues versus lovers). It is also central to SSR, where mental state inferences are modulated by relational roles (e.g., a critical comment from a mentor versus a stranger implies different states). For SDP, predictions of future behaviors and counterfactual reasoning are shaped by relational norms and history (e.g., conflict resolution strategies differ between siblings and business rivals).

SIV-Bench comprises 2,792 video clips sourced from TikTok and YouTube. Its video corpus ensures robust evaluation by encompassing **diverse genres** (such as daily life recordings, movie clips, sports footage, and animation), **varied presentation styles** (including first-person, third-person, phone calls, and solo multi-role sketches) and content reflecting **multiple linguistic and cultural backgrounds** (like English, Japanese, and Spanish). A dedicated question-answer (QA) pipeline, combining large language models with human verification, produces 8,728 high-quality multiple-choice questions (MCQs). These MCQs are designed to target specific facets within our three core assessment dimensions. Furthermore, to systematically assess the impact of linguistic cues, SIV-Bench incorporates audio tracks and provides videos under three distinct subtitle conditions: the original version (origin), a version with transcribed and translated dialogue added (+sub), and a version with all original on-screen text removed (-sub), as illustrated in Figure 4.

In the experiments, we evaluate leading commercial(e.g., Gemini-2.5-Pro (Doshi, 2025), o4-mini (OpenAI, 2025)), open-source (e.g., InternVL3 (Zhu et al., 2025a), Qwen2.5-VL (Bai et al., 2025)), and specialized video models (e.g., LLaVA-OneVision (Li et al., 2024a), LLaVA-Video (Zhang et al., 2024)). The results show clear performance differences across the three dimensions. Models perform relatively well in SSU but struggle with SSR, where Relation Inference proves especially challenging. Common failure patterns include confusion between primary and secondary relationships, misleading

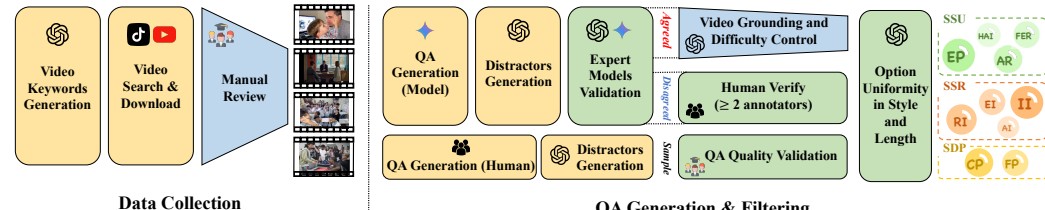

Figure 2: The SIV-Bench construction pipeline, detailing the data collection process (left), and the QA generation & filtering process (right) with human-LLM collaboration. In the diagram, ▨ blocks indicate content (like keywords, video and QA) generation steps, ▨ blocks represent validation or modification stages, and ▨ trapezoids signify a filtering and removal phase.

contextual cues, weak commonsense reasoning, and missed perceptual details. SDP also poses difficulties, though top models handle counterfactuals relatively well. Our study also underscores the general importance of linguistic cues, as transcribed subtitles consistently aid overall comprehension while their removal typically hinders performance.

Our key contributions are as follows: **1)** We propose a novel analytical framework that structurally decomposes the complex task of multimodal social interaction understanding and reasoning into three core, interrelated dimensions, each further detailed into fine-grained sub-tasks. **2)** We introduce SIV-Bench, a new video benchmark specifically curated for the analysis and comprehension of complex real-world social interactions. SIV-Bench comprises 2,792 real-world video clips representing 14 distinct social relationship types, and features 8,792 high-quality question-answer pairs generated through a human-LLM collaborative pipeline. **3)** Our comprehensive experiments on diverse MLLMs reveal the limitations in their current capacity for deep human social understanding. These findings offer crucial insights to direct future research toward advancing artificial social intelligence.

## 2 SIV-BENCH

This section details SIV-Bench, our novel video benchmark developed to evaluate MLLMs' capabilities in understanding and reasoning within social interaction scenarios. Figure 1 offers some illustrative examples from SIV-Bench. The video corpus features interactions curated across 4 primary relational models, further detailed into 14 specific social relations, to ensure comprehensive coverage of real-world scenarios. The associated QA pairs are meticulously designed to probe model capabilities across 3 major task categories and 10 fine-grained aspects, targeting understanding, reasoning, and prediction in social contexts. Figure 2 shows the construction pipeline of SIV-Bench, including video collection (Section 2.1), QA generation and filtering (Section 2.2). Finally, to contextualize its contributions, SIV-Bench is also compared with other relevant benchmarks and datasets (Section 2.3).

### 2.1 VIDEO COLLECTION

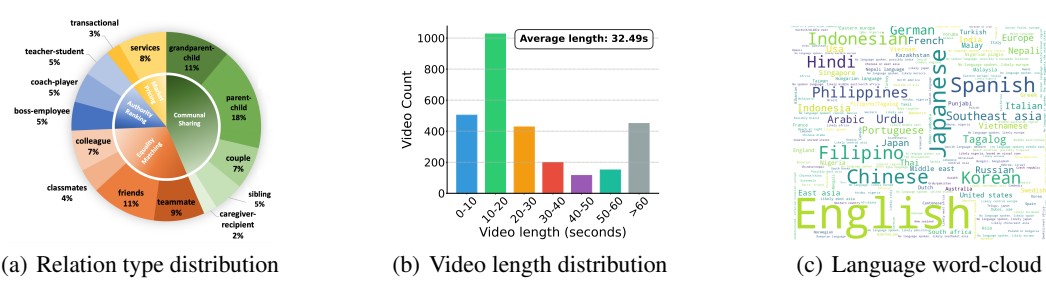

(a) Relation type distribution (b) Video length distribution (c) Language word-cloud

Figure 3: Video statistics for SIV-Bench: (a) Distribution of social relation types. (b) Distribution of video lengths, with an average of 32.49s. (c) Word cloud illustrating language diversity.

Firstly, we utilize GPT-4o-mini (Hurst et al., 2024) to generate comprehensive search keywords for each of the 14 relationship types (the word-clouds are shown in Figure 7), specifically including

terms associated with varying degrees of intimacy, both positive (e.g., "love", "encourage") and negative (e.g., "conflict", "fight"). Leveraging these keywords, we conduct targeted searches and download initial video candidates from TikTok and YouTube platforms using Python libraries such as `TikTokApi` and `yt-dlp`, yielding approximately 5000 raw video clips. Each video is then manually reviewed by the authors to ensure it contains clearly observable and meaningful social interactions. Videos are excluded if they do not depict clear social interaction (e.g., a vlogger speaking directly to the camera without interacting with others), if the interaction context does not fit within a set of well-defined interpersonal scenarios (e.g., an interview setting with scripted dialogue), or if the dominant interaction is difficult to identify due to the presence of multiple overlapping social dynamics (e.g., a large multi-generational family posing for a group photo). These criteria are designed to ensure that each included video primarily features one interpretable and coherent type of interpersonal interaction, allowing for more consistent analysis of social behaviors across the dataset.

The final SIV-Bench comprises **2,792** curated video clips, with statistics shown in Figure 3. The collection showcases a rich diversity in social relationships (Figure 3(a)). Communal Sharing interactions are the most represented category, a distribution intentionally preserved to reflect the naturalistic prevalence and psychological centrality of these relationships in daily life (Simão & Seibt, 2014; Kameda et al., 2005). The other three relational models are also well represented, ensuring broad interpersonal coverage. In terms of duration (Figure 3(b)), clips average 32.49 seconds.

While most clips are 10–20 seconds, the dataset spans a wide distribution, including many short clips (under 10 seconds) and a significant number over 60 seconds. English predominates (Figure 3(c)), but SIV-Bench also includes Spanish, Filipino, Korean, and Japanese, adding multicultural diversity (Triandis, 1989). Additional details on video genre and style are in Appendix B.2.

To evaluate how different forms of textual information affect MLLMs' understanding of social interactions, we implement specific subtitle processing methods (Figure 4). Many original videos ('Origin') contain embedded on-screen text that often serves as scene descriptors or keywords (e.g., the 'Buy and Sell' text overlay). To focus on visual and auditory cues, we create a '-Subtitle' version by removing such original textual overlays using

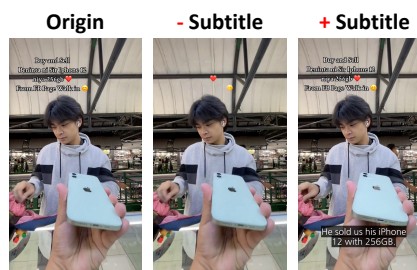

Figure 4: Illustration of the three subtitle conditions applied in SIV-Bench: 'Origin', '-Subtitle' and '+Subtitle' .

`video-subtitle-remover` (YaoFANGUK, 2025). Conversely, to provide full access to spoken dialogue, we generate a '+Subtitle' version. We employ Whisper-large-v3 (Radford et al., 2022) for audio transcription of the dialogue, and then use GPT to translate these transcriptions into English, ensuring consistent and high-quality subtitles (e.g., 'He sold us his iPhone 12 with 256GB'.).

## 2.2 QA COMPOSING

Our Question-Answer (QA) composition process begins with generating an initial set of diverse QA pairs for each video, leveraging the capabilities of Gemini-2.0-Flash (Google, 2025b) which supports full video input. The prompts are carefully designed to elicit questions that span a broad range about social interactions, forming the basis for our SSU, SSR, and SDP evaluation dimensions. SSU focuses on descriptive observations of scenes, actions, and attributes; SSR targets inferences about intents, emotions, and relations; and SDP involves factual/counterfactual predictions. This stage typically yielded an average of 10 versatile QA pairs per video, designed to cover these core dimensions. Then we employ GPT-4o-mini to generate four distractors for each QA pair. These distractors are crafted to be contextually relevant to the question while remaining clearly distinguishable from the correct answer. We find that separating QA creation and distractor generation into two stages significantly improves quality and reduces ambiguity over generating all options simultaneously by a single model.

For quality assurance and to establish a gold standard, all generated QAs then undergo a rigorous consensus validation step (Chen et al., 2023a;b; Amiri-Margavi et al., 2024). We utilize three expert models (Gemini-2.0-Flash, Gemini-2.0-Pro, and GPT-4o-mini) to independently answer all questions. The QAs on which these models agree constitute our primary high-confidence set (Subset 1). The

remaining QAs, where model agreement is not achieved, are flagged for further review or specialized handling (Subset 2). The following parts detail the subsequent processing strategies for each subset:

**Subset 1:** For Subset 1, which comprises QAs where expert models initially reached consensus (potentially indicating a need to increase challenge or verify video dependency), we implement a two-step filtering process. First, GPT-4o-mini is utilized to answer these questions without any video input. Questions that are answered correctly under this condition are removed. While this is a stringent criterion, as the model might occasionally guess correctly without sufficient information, it serves as an effective filter to ensure the questions necessitate video-based understanding, a worthwhile trade-off given our large initial pool of generated QAs. Second, the remaining questions are then scored for difficulty by GPT-4o-mini on a 1-to-5 scale. From this pool, we select the top 1-2 highest-scoring (i.e., most difficult) questions per video. Following these automated filtering stages, the authors conduct a final cross-validation review of these selected QAs to ensure their overall quality, relevance, and appropriate difficulty level. This multi-stage process ultimately yields 3,273 high-quality QAs.

**Subset 2:** We recruit 20 human annotators to verify non-consensus items, with each QA pair reviewed by at least two people. Only questions for which all reviewers independently agree are retained as ground truth, yielding 3,096 QAs. The annotators also generate novel, challenging questions with corresponding answers, following instructions targeting diverse abilities across the 10 fine-grained tasks. These questions are deliberately difficult, requiring detailed understanding or complex reasoning. Appendix C.3 details the human annotation process, including annotator demographics, guidelines, and the custom interface. For the 2,359 new human-generated QAs, plausible distractors are created using GPT-4o-mini.

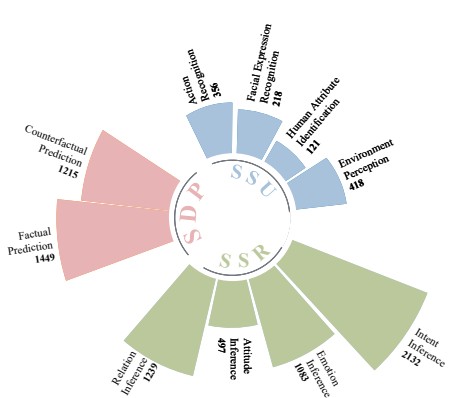

Figure 5: Detailed statistics of Question-Answer pairs in SIV-Bench, showing the distribution across the 10 fine-grained sub-tasks.

Then, all curated QA pairs undergo a final automated refinement stage, where GPT-4o-mini processes both correct answers and distractors. This step standardizes linguistic style and option length, thereby minimizing superficial cues that models might exploit, improving test reliability, and reducing variation introduced by annotators or earlier generation steps. Full prompts and implementation details are provided in Appendix C.1 and C.2.

Finally, SIV-Bench contains a total of **8,728** QA pairs. Their distribution across core assessment dimensions and fine-grained tasks is illustrated in Figure 5. SRR is the largest category (4,951 QAs), followed by SDP (2,664) and SSU (1,113). Sub-tasks such as Intent Inference (2,132) and Factual Prediction (1,449) are especially concentrated, enabling rich evaluation. Appendix C.4 (Figure 17) provides detailed statistics confirming structural balance and diversity (e.g., option length, answer distribution, question complexity, interrogatives). Appendix C.5 further shows high linguistic diversity and robustness against template-based cues.

## 2.3 COMPARISON WITH PREVIOUS DATASETS & BENCHMARKS

Table 1 highlights the key differences between SIV-Bench and previous datasets and benchmarks. Prior datasets on social interaction, such as DialogRE (Yu et al., 2020) (text-only), PIPA (Sun et al., 2017) (image-only), and ViSR (Liu et al., 2019), are limited to relation recognition, single modalities, and lack task diversity and scalability due to manual annotation. Currently, most general video understanding benchmarks (like VideoVista (Li et al., 2024c) and MVBench (Li et al., 2024b)) lack a focus on social interaction; others (like MLVU (Zhou et al., 2024) and Video-Bench (Ning et al., 2023)), while touching upon it, do not fully cover social relations. Even Social-IQ 2.0 (Wilf et al., 2023), which concentrates on this area, has limitations in task diversity and dynamic reasoning. In contrast, SIV-Bench is built on original data, combines manual and automatic annotations, and is one of the few benchmarks to provide subtitle and audio information, thereby supporting richer multimodal social reasoning.

Table 1: Comparison of various benchmarks. It includes several aspects: total number of items (**#Items**, representing the number of videos, dialogues, images, etc.), number of QA pairs (**#QAs**), annotation method (**Anno.**, M/A means manually/automatic manner), and the types of tasks included (*Task Types*: **SU** for Scene Understanding, **SR** for State Reasoning, **DP** for Dynamics Prediction). Note that *Task Types* here refer to general task categories, which include both social and physical scenarios. The table also shows whether each benchmark includes **Multi-Person Interaction**, covers **Various Relations**, is based on newly collected data (**Original Collection**), and provides subtitle/audio information (**S.A.**)

| Benchmark | #Items | #QAs | Anno. | Task Types | | | Multi-Person Interaction | Various Relations | Original Collection | S.A. |
|---|---|---|---|---|---|---|---|---|---|---|
| | | | | SU | SR | DP | | | | |
| *Social Relation Inference* | | | | | | | | | | |
| DialogRE (Yu et al., 2020) | 1,788 | - | M | ✓ | ✓ | ✗ | ✓ | ✓ | ✗ | - |
| PIPA (Sun et al., 2017) | 37,107 | - | M | ✗ | ✓ | ✗ | ✓ | ✓ | ✗ | - |
| ViSR (Liu et al., 2019) | 8,000 | - | M | ✗ | ✓ | ✗ | ✓ | ✓ | ✓ | ✗ |
| *General Video Understanding and Reasoning* | | | | | | | | | | |
| VideoMME (Fu et al., 2024) | 900 | 2,700 | M | ✓ | ✓ | ✗ | ✓ | ✗ | ✓ | ✓ |
| MLVU (Zhou et al., 2024) | 1,730 | 3,102 | M | ✓ | ✗ | ✓ | ✓ | ✗ | ✗ | ✗ |
| VideoVista (Li et al., 2024c) | 894 | 24,906 | A | ✓ | ✗ | ✓ | ✗ | ✗ | ✗ | ✓ |
| Social-IQ 2.0 (Wilf et al., 2023) | 1,000 | 6,000 | M | ✓ | ✓ | ✗ | ✓ | ✓ | ✓ | ✗ |
| Social Genome (Mathur et al., 2025) | 272 | 1,486 | M | ✓ | ✓ | ✗ | ✓ | ✗ | ✗ | ✓ |
| Perception Test (Patraucean et al., 2023) | 11,620 | 38,000 | M | ✓ | ✗ | ✓ | ✗ | ✗ | ✓ | ✗ |
| MVBench (Li et al., 2024b) | 3,641 | 4,000 | A | ✓ | ✗ | ✓ | ✗ | ✗ | ✗ | ✗ |
| Video-Bench (Ning et al., 2023) | 5,917 | 17,036 | A&M | ✓ | ✓ | ✗ | ✓ | ✗ | ✗ | ✗ |
| EgoSchema (Mangalam et al., 2023) | 5,063 | 5,063 | A&M | ✓ | ✓ | ✗ | ✓ | ✗ | ✗ | ✗ |
| SIV-Bench | 2,792 | 8,728 | A&M | ✓ | ✓ | ✓ | ✓ | ✓ | ✓ | ✓ |

## 3 EXPERIMENTS

### 3.1 SETTINGS

We evaluate a diverse set of closed- and open-source MLLMs on SIV-Bench, including Gemini-2.0/2.5-Flash (Google, 2025b;a), Gemini-2.5-Pro (Doshi, 2025), GPT-4o (Hurst et al., 2024), o4-mini (OpenAI, 2025), Qwen2.5-VL-7B/72B-Instruct (Bai et al., 2025), mPLUG-Owl3 (Ye et al., 2024), InternVL3-8B/78B (Zhu et al., 2025a), LLaVA-OneVision (Li et al., 2024a), and LLaVA-Video (Zhang et al., 2024). Evaluations are conducted using VLMEvalKit (Duan et al., 2024), with models generating responses under their default settings. Video-capable models (e.g., Gemini) receive raw videos, while image-only models (e.g., o4-mini) are given 16 uniformly sampled frames.

We use a standardized prompt (Figure 18) across all models, instructing them to return both the answer letter (e.g., 'A.') and the full answer text. However, we observe variations in model compliance: for instance, some models (e.g., InternVL-8B and mPLUG-Owl3) occasionally tend to output only the letter, whereas others (e.g., LLaVA-Video) may provide only the text. To robustly handle these diverse output formats and ensure accurate answer parsing, we implement a two-stage matching procedure. First, we check the model's output for a correctly formatted option letter. If absent, we use a similarity-based comparison between the model's raw output and the full text of all answer options to identify its selection. Model performance is then measured by the accuracy of these inferred responses, as shown in Table 2. The rest of this section provides a detailed analysis.

### 3.2 MAIN RESULTS

**Overall Performance.** The evaluation of various models on SIV-Bench reveals distinct performance tiers, with closed-source models generally setting the current state-of-the-art. Notably, Gemini-2.5-Pro emerges as the top-performing model, achieving the highest overall accuracy of 76.50% when provided with subtitles. Other prominent closed-source models, such as GPT-4o and Gemini-2.5-Flash, also demonstrate strong overall performance, consistently scoring above 73%. These results indicate the advanced capabilities of leading proprietary systems.

Following these, open-source Multi-modal Large Language Models (MLLMs) show considerable capabilities, although a performance gap still exists. Among them, models with a larger number of parameters exhibit significantly better results; for example, Qwen2.5-VL-72B-Instruct achieves an overall score of 74.77% with subtitles, markedly outperforming its 7B counterpart (65.13%

Table 2: Evaluation results of MLLMs on SIV-Bench. Scores are reported for SSU, SSR, SDP, and Overall performance, detailing the impact of subtitle conditions: 'origin,' '+sub' (added transcribed dialogue), and '-sub' (subtitles removed). All scores are presented as accuracy percentages (%).

| Models | Params | Social Scene Understanding | | | Social State Reasoning | | | Social Dynamics Prediction | | | Overall | | |
|---|---|---|---|---|---|---|---|---|---|---|---|---|---|
| | | origin | + sub | - sub | origin | + sub | - sub | origin | + sub | - sub | origin | + sub | - sub |
| *Open-source MLLMs* | | | | | | | | | | | | | |
| mPLUG-Owl3 | 7B | 66.32 | 66.21 | 66.46 | 62.36 | 62.19 | 61.33 | 65.19 | 66.30 | 65.22 | 63.79 | 64.01 | 63.22 |
| LLaVA-OneVision | 7B | 62.00 | 62.13 | 62.55 | 63.72 | 64.78 | 61.66 | 64.87 | 65.32 | 62.14 | 63.73 | 64.40 | 62.14 |
| LLaVA-Video | 7B | 68.89 | 69.13 | 69.10 | 62.08 | 61.37 | 60.09 | 63.50 | 63.84 | 62.46 | 63.18 | 63.75 | 61.66 |
| Qwen2.5-VL-7B-Instruct | 7B | 69.51 | 69.30 | 68.89 | 62.65 | 61.84 | 61.04 | 64.18 | 64.89 | 63.90 | 65.01 | 65.13 | 63.53 |
| InternVL3-8B | 8B | 73.02 | 72.58 | 72.81 | 62.72 | 63.06 | 61.20 | 65.33 | 65.95 | 65.25 | 66.14 | 66.28 | 65.35 |
| Qwen2.5-VL-72B-Instruct | 72B | 84.83 | 85.15 | 83.46 | 70.23 | 70.54 | 69.58 | 74.88 | 74.49 | 74.11 | 74.25 | 74.77 | 73.54 |
| InternVL3-78B | 78B | 82.16 | 83.54 | 82.35 | 67.45 | 67.91 | 66.87 | 73.14 | 73.46 | 72.21 | 72.16 | 72.70 | 71.56 |
| *Closed-source MLLMs* | | | | | | | | | | | | | |
| o4-mini | - | 86.77 | 86.90 | 86.33 | 66.87 | 67.25 | 66.10 | 72.86 | 72.45 | 72.11 | 72.30 | 72.43 | 71.59 |
| GPT-4o | - | 86.94 | 87.34 | 86.29 | 68.28 | 68.58 | 67.73 | 74.82 | 75.80 | 74.05 | 73.76 | 74.29 | 73.12 |
| Gemini-2.0-Flash | - | 86.54 | 86.35 | 86.46 | 67.73 | 68.07 | 66.49 | 73.14 | 73.79 | 72.42 | 72.75 | 73.27 | 71.65 |
| Gemini-2.5-Flash | - | 88.56 | 88.84 | 87.32 | 68.12 | 69.09 | 67.25 | 74.67 | 74.97 | 73.05 | 73.67 | 73.82 | 72.53 |
| Gemini-2.5-Pro | - | **90.67** | **90.88** | **90.59** | **71.44** | **71.78** | **70.20** | **75.28** | **75.96** | **74.27** | **76.03** | **76.50** | **75.14** |

with subtitles), even superior to the closed-source models except for Gemini-2.5-Pro. Similarly, InternVL-78B performs substantially better than InternVL-8B. While these larger open-source models demonstrate strong potential and achieve commendable overall scores, there remains room for improvement to match the leading closed-source alternatives.

**Subtitle and Audio Influence.** Our analysis reveals that while linguistic cues benefit MLLMs, their impact is nuanced. To quantify the audio channel's contribution, we conducted an ablation study on Gemini models (Table 3). Removing audio from original videos consistently degrades performance, confirming that auditory cues like prosody and tone contribute to understanding. Conversely, the negligible impact of

Table 3: Audio ablation study on Accuracy (%).

| Model | Condition | Accuracy |
|---|---|---|
| Gemini-2.0-Flash | origin w/ vs. w/o audio | 72.75 → 70.81 |
| | +sub w/ vs. w/o audio | 73.27 → 73.19 |
| Gemini-2.5-Flash | origin w/ vs. w/o audio | 73.67 → 71.95 |
| | +sub w/ vs. w/o audio | 73.82 → 73.71 |

removing audio from +sub videos validates that our high-quality subtitles effectively convey the necessary linguistic information for models without native audio processing. More broadly, analyzing subtitle modifications across all models (Table 4) shows that SSU, relying on visual cues, is least affected. In contrast, the more inferential SSR and SDP tasks show greater sensitivity. Removing all on-screen text (−sub) most significantly impacts SSR (a relative drop of -1.176%), underscoring the critical role of linguistic information in reasoning about complex social interactions.

**Task-Specific Performance.** SSU, focused on recognizing visible elements, is typically the easiest, with models achieving their highest scores here. However, this isn't always the case—for instance, LLaVA-OneVision sometimes performs better on SSR or SDP. In SSU, stronger models—especially large-scale ones, whether closed-source or high-parameter open-source—consistently outperform smaller mod-

Table 4: Average relative performance change (%) of all models when adding ('+sub') or removing ('-sub') subtitles, compared to the 'origin'.

| Condition | SSU (%) | SSR (%) | SDP (%) | Overall (%) |
|---|---|---|---|---|
| + sub | 0.178 | 0.234 | **0.482** | 0.447 |
| - sub | -0.300 | **-1.176** | -0.878 | -0.878 |

els. This performance gap, particularly pronounced in this foundational visual task, likely reflects the superior perceptual capabilities of larger models due to greater capacity and more extensive training (Alabdulmohsin et al., 2022). Following SSU, SDP tends to be the next most tractable, as it involves reasoning about interactions and predicting outcomes. SSR, which requires inferring unobservable social states like emotions and intentions, remains the most difficult. Even the top models score lowest here, underscoring the challenge of capturing implicit psychological dimensions.

To further break down model capabilities, a radar chart (Figure 6) visualizes performance across 10 fine-grained tasks in original videos (with '+sub' and '-sub' detailed in Appendix D.2). Stronger models, particularly closed-source ones like the Gemini series and GPT-4o, form the outer ring, clearly outperforming smaller open-source models. In SSU, the performance gap is especially evident in Action Recognition (AR) and Facial Expression Recognition (FER), reflecting the stronger models' ability to capture subtle visual cues. SSR presents greater challenges: while models perform

moderately on Intent Inference (II) and Emotion Inference (EI), they struggle with more abstract tasks like Relation Inference (RI), which often mark the lowest points on the radar. This illustrates the difficulty of moving from surface-level cues to deeper social reasoning. In SDP, performance improves slightly, likely because tasks such as Factual Prediction (FP) and Counterfactual Prediction (CP) tap into social commonsense learned from language data. Most models—except a few like LLaVA-OneVision—perform better on CP than FP, possibly due to the more explicit reasoning cues provided by hypothetical framing. Some failure cases are illustrated in Figure 20, 21, 22. To rigorously assess the upper bounds of model capability and rule out benchmark saturation, we further conducted an analysis on **SIV-Bench-Hard**, a curated subset of 200 challenging human-authored questions. As detailed in Appendix E, this evaluation establishes a human baseline and reveals a substantial human-AI performance gap regarding both accuracy and reasoning quality.

### 3.3 Failure Pattern Analysis of Relation Inference

Accurate Relation Inference (RI) is crucial for understanding social interactions. The difficulty lies in synthesizing diverse cues—scene context, human appearance, verbal and non-verbal expressions, physical proximity, and interaction dynamics—all of which contribute to subtle and implicit relational patterns. Our fine-grained analysis (Figure 6) highlights RI as a consistent performance bottleneck. Given its significance and the persistent challenges across models, we conduct a deeper qualitative analysis to examine common failure patterns:

**Lack of Primary–Secondary Relation Differentiation.** In multi-relational scenarios, MLLMs sometimes identify a valid but secondary relationship rather than the most salient one. For example, in a video where employees celebrate their boss's birthday, the model labels the interaction as <colleagues>, overlooking the hierarchical bond that defines the scene. Similarly, when a coach angrily reprimands players, the model assigns <teammates>, missing the coach–athlete authority structure. These errors highlight limitations in relational prioritization and in discerning the most significant social bond within complex interpersonal contexts.

**Scene and Human-Induced Misleading Cues.** MLLMs are easily misled by both scene settings and human behaviors. They may over-rely on environmental cues, assigning stereotypical roles based on location without validating actual interactions. For instance, in a classroom setting where two travel agency employees co-present, it incorrectly classifies them as <teacher-student> instead of <colleagues>. Similarly, human language can mislead the model: a couple jokingly referring to each other as "best brother" is misclassified as <siblings>, despite strong visual evidence—such as different ethnicities and intimate gestures—indicating a romantic relationship.

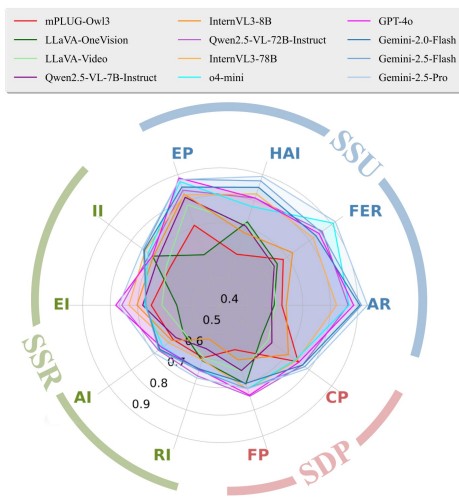

Figure 6: Radar chart of MLLM performance across the 10 fine-grained SIV-Bench sub-tasks.

**Deficiency in Commonsense Reasoning.** In several instances, the model's decisions are inconsistent with basic commonsense knowledge. Two <friends> wearing different sports team uniforms but showing friendly behavior are misclassified as <teammates>, disregarding the fact that such uniforms typically signify competition rather than collaboration. Similarly, a store owner photographing a bustling crowd is mistaken for a colleague of the customers, whereas commonsense suggests a <transactional> relationship. These mistakes highlight the model's limited ability to integrate external knowledge about social conventions and situational functions, emphasizing a shortfall in commonsense reasoning and real-world social understanding.

**Ignoring Key Perceptual Details.** Although MLLMs can capture many surface features, they will overlook subtle but critical details necessary for correct role recognition. Age differences and facial features crucial to distinguishing between <siblings> and <parent-child> relationships are sometimes

ignored. Additionally, important textual clues embedded in subtitles are sometimes overlooked, such as the line "POV: you work with your coworker," which results in <colleague> not being correctly assigned. These errors suggest insufficient fine-grained perception, selective attention to critical multimodal information, and a failure to integrate minor yet decisive cues into final role judgments.

To complement this qualitative analysis with quantitative evidence, we provide a detailed performance breakdown across the 14 relation types and an aggregated confusion matrix in Appendix D.2.

## 4 RELATED WORKS

**Video Benchmarks for MLLMs.** The evaluation of rapidly advancing Multimodal Large Language Models (MLLMs) in the video domain relies on diverse benchmarks that collectively incorporate a rich set of features for comprehensive model assessment. Many, such as MLVU (Zhou et al., 2024), video-MME (Fu et al., 2024), MVbench (Li et al., 2024b), MMBench-Video (Fang et al., 2024), and Video-Bench (Ning et al., 2023), offer multiple tasks—ranging from question answering and summarization to complex reasoning—across diverse video fields including movies, egocentric recordings, and general web content. This variety aims to assess a broad spectrum of capabilities, from foundational visual understanding and detail analysis (as targeted by Perception Test (Patraucean et al., 2023)) (explored by VideoVista (Li et al., 2024c)) to higher-level comprehension and decision-making. A significant trend within these evaluations is the robust focus on long-form video understanding, with benchmarks like CINEPILE (Rawal et al., 2024), Egoschema (Mangalam et al., 2023), and LVbench (Wang et al., 2024b) specifically challenging models on temporal reasoning and extended context retention, while others like TempCompass (Liu et al., 2024) provide specialized assessments of critical nuanced aspects such as detailed temporal dynamics. While these benchmarks thoroughly assess diverse video comprehension capabilities, SIV-Bench offers a more specialized focus, concentrating on the intricate understanding and reasoning required to interpret dynamic social interactions among multiple people.

**Evaluating Social Intelligence in AI Systems.** The pursuit of artificial social intelligence is inherently multifaceted, with research progressing from early dataset construction to contemporary evaluations of large language models (LLMs). Initial efforts typically offered valuable yet modality-constrained or component-specific resources, targeting areas such as action recognition (e.g., Kinetics (Kay et al., 2017), UCF101 (Soomro et al., 2012)), facial expression and contextual emotion inference (e.g., AffectNet (Mollahosseini et al., 2017), MELD (Poria et al., 2018)), and social relation recognition from static images or text (e.g., PIPA (Sun et al., 2017), DialogRE (Yu et al., 2020)). As the field has evolved, benchmarks like Social-IQ 2.0 (Wilf et al., 2023) have expanded the focus toward interpersonal reasoning and social commonsense. With the rise of LLMs, more specialized evaluations have emerged. SMILE (Hyun et al., 2024) introduces video laugh reasoning to interpret the rationale behind laughter. ToMATO (Shinoda et al., 2025) probes Theory of Mind through role-play-based interactions. SOTOPIA (Zhou et al., 2023) and its extension SOTOPIA-$\pi$ (Wang et al., 2024a) introduce interactive environments for developing agentive social skills. EgoSocialArena (Hou et al., 2024) offers a first-person perspective benchmark targeting both cognitive and behavioral intelligence. Social Genome (Mathur et al., 2025) provides multimodal, grounded traces of social reasoning from video data, while MM-SOC (Jin et al., 2024) evaluates MLLMs on multimodal social media content. Despite these advances, most existing benchmarks focus on isolated competencies. A comprehensive evaluation framework that holistically assesses MLLMs' understanding of complex social interactions—spanning perception, social state reasoning, and dynamics prediction in real-world video contexts—remains lacking. SIV-Bench is introduced to fill this gap.

## 5 CONCLUSION

In this work, we introduce SIV-Bench, a novel video benchmark designed to rigorously evaluate Multimodal Large Language Models' capabilities in understanding and reasoning about complex social interactions. Featuring 2,792 diverse videos categorized by foundational social relationship models and 8,792 meticulously generated question-answer pairs, SIV-Bench structures evaluation around three core dimensions: Social Scene Understanding (SSU), Social State Reasoning (SSR), and Social Dynamics Prediction (SDP), further detailed into ten fine-grained sub-tasks. Our comprehensive experiments reveal that while leading MLLMs adeptly handle SSU, they still struggle with SDP and

SSR, where Relation Inference emerges as an acute bottleneck. Moreover, our study underscores the critical impact of transcribed dialogue on comprehending complex social interactions.

Although SIV-Bench provides a strong evaluation framework, its current scale suggests room for future expansion into a larger training corpus with broader social contexts and more diverse content. The multiple-choice format also offers a basis for future extensions into interactive or generative evaluation. By identifying current model strengths and weaknesses, SIV-Bench offers timely insights and a foundation for advancing socially intelligent AI.

## ETHICS STATEMENT

The authors have read and adhere to the ICLR Code of Ethics. This work was conducted with careful consideration for the ethical implications of creating and distributing a benchmark based on public social media data. We outline our approach to key ethical issues below.

**Data Sourcing and Licensing.** The videos in SIV-Bench were sourced from publicly available content on TikTok and YouTube. We acknowledge that the redistribution of video files raises complex issues regarding platform Terms of Service and creators' rights. To address this, the final public release of SIV-Bench will not distribute video files directly. Instead, we will provide a download script containing only the video URLs, timestamps, and our corresponding annotations. This requires end-users to fetch the videos directly from their original platforms, a standard practice that respects copyright and aligns with platform policies.

**Human Subjects and Privacy.** The dataset contains videos of identifiable individuals who have made their content publicly available. During the collection phase, we performed a manual filtering process to exclude content that appeared overly private, sensitive, or depicted minors in potentially vulnerable situations. We acknowledge that de-identification methods, such as face-blurring, represent a higher standard for privacy protection. While this was not implemented in the current version due to the significant manual effort required, we recognize its importance. We commit to including a clear discussion of this limitation in the public dataset documentation to ensure that future users are fully aware of the data's nature. This transparency is crucial for promoting responsible downstream use.

**Human Annotation Process.** All 20 human annotators involved in this project were treated ethically. Before participation, each annotator was provided with a detailed informed consent form that outlined the study's purpose, the nature of the annotation tasks, the compensation structure ($4 USD/hour, meeting or exceeding standard local wages for such work), and how their anonymized data would be used. The tasks were designed to be minimal-risk, and all data collected from annotators was handled anonymously throughout the research process.

**Intended Use and Broader Impact.** SIV-Bench is intended as a research tool to transparently assess the capabilities and limitations of MLLMs in the social domain. By highlighting specific weaknesses, such as in Relation Inference, we hope to steer the community toward developing more robust and socially aware AI. We recognize that technologies for social understanding have dual-use potential. We encourage the community to use this benchmark to proactively consider these challenges and develop necessary safeguards alongside continued innovation.

## REPRODUCIBILITY STATEMENT

To ensure the reproducibility of our research, we have provided a comprehensive suite of resources. The complete source code for our data processing and model evaluation is available as an anonymized submission in the supplementary materials. Due to file size limitations, this submission includes a representative subset of the video data. We are committed to releasing the full SIV-Bench dataset, including all video download scripts and corresponding annotations, along with the complete codebase, upon the paper's formal acceptance, following the ethical guidelines outlined in our Ethics Statement.

Our pipeline for data collection is detailed in Section 2.1. The human-LLM collaborative pipeline for generating and filtering question-answer (QA) pairs is described in Section 2.2, with the specific prompts and filtering strategies documented in Appendix C.1 and C.2. A detailed statistical breakdown of the SIV-Bench dataset, including the distribution of relation types, video lengths, and languages,

can be found in Section 2.1 (Figure 3), with further analysis of the QA pairs provided in Appendix C.4 (Figure 17).

Our experimental setup for evaluating Multimodal Large Language Models (MLLMs), including the models tested and the hardware used, is described in Section 3.1. All standardized prompts used for model evaluation are fully documented in Appendix D.1 to facilitate the direct replication of our results. The default inference parameters for all models were adopted from the VLMEvalKit framework, as noted in Appendix D.1, to ensure a fair and consistent comparison.

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

## A  STATEMENT ON THE USE OF LARGE LANGUAGE MODELS

The LLM was utilized exclusively to aid and polish the writing, including for tasks such as improving grammar and clarity, refining sentence structure, and ensuring stylistic consistency.

## B  VIDEO DETAILS

### B.1  VIDEO COLLECTION

To compile a rich and diverse video corpus for SIV-Bench that reflects a wide array of social interactions, we first identify 14 key social relationship types (e.g., parent-child, friends, colleagues, service interactions). For each of these types, we employ GPT-4o mini to generate a comprehensive set of search keywords. This strategy is designed to ensure broad coverage in our video collection from platforms like YouTube and TikTok, capturing various interaction categories (such as cooperation, conflict, and support), forms (including verbal discussions, non-verbal expressions, and shared activities), and differing levels of intimacy or formality inherent to each relationship. Figure 7 visualizes these tailored keyword sets as word clouds for each of the 14 relationship types, illustrating the breadth of concepts targeted. This targeted yet diverse keyword generation is crucial for assembling a video dataset that truly represents the complexity of human social life.

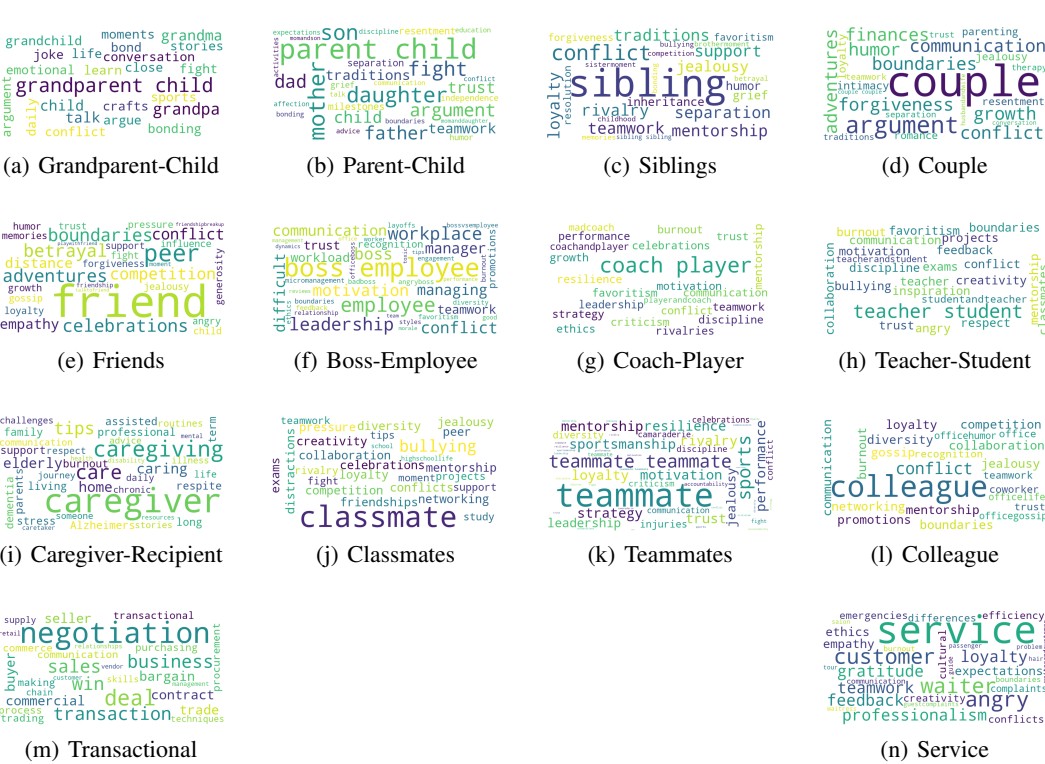

| (a) Grandparent-Child | (b) Parent-Child | (c) Siblings | (d) Couple |
| (e) Friends | (f) Boss-Employee | (g) Coach-Player | (h) Teacher-Student |
| (i) Caregiver-Recipient | (j) Classmates | (k) Teammates | (l) Colleague |
| (m) Transactional | | | (n) Service |

Figure 7: Word clouds of GPT-generated keywords used for sourcing videos across 14 distinct social relationship types (e.g., (a) Grandparent-Child, (b) Parent-Child, (e) Friends, (f) Boss-Employee). Keywords were designed to capture diverse interaction scenarios, forms, and intimacy levels from platforms like YouTube and TikTok.

### B.2  VIDEO DIVERSITY

The SIV-Bench video corpus is intentionally diverse in both subject matter and presentation to ensure comprehensive model evaluation. This genre diversity, illustrated in Figure 8, includes a wide array of video types such as candid Daily Life recordings, scripted Movie Clips, dynamic Sports Replays,

illustrative Commercials, and non-photorealistic Animated Videos. Complementing this, Figure 9 showcases varied visual presentation styles, ranging from traditional Third-Person perspectives and immersive First-Person (POV) footage to interactions depicted via Phone Call interfaces and creative Solo Multi-Role Sketches. This multifaceted diversity in video content and form provides a robust and varied testbed.

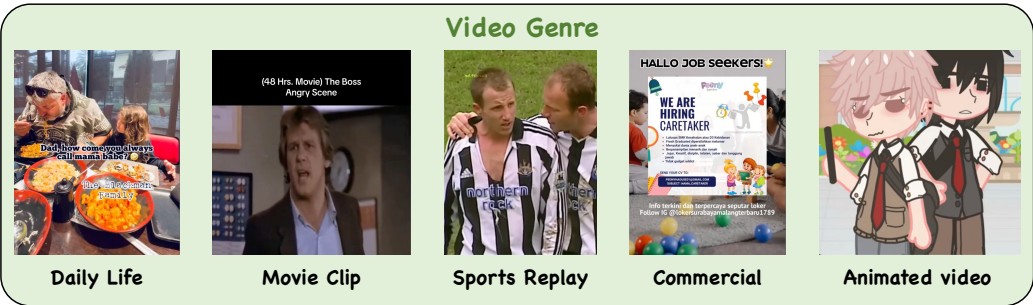

Figure 8: Examples illustrating the diverse video genres in SIV-Bench, including (from left to right) Daily Life recordings, Movie Clips, Sports Replays, Commercials, and Animated Videos, ensuring a broad range of social interaction contexts.

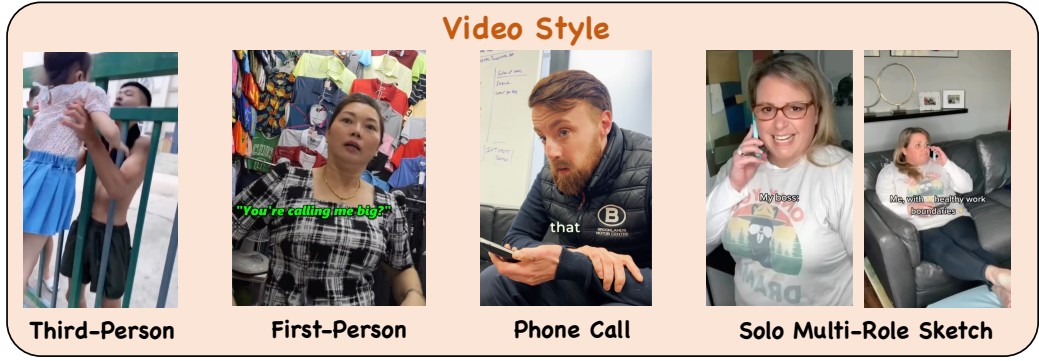

Figure 9: Examples of diverse video presentation styles featured in SIV-Bench, including (from left to right) Third-Person views, First-Person perspectives, Phone Call interfaces, and Solo Multi-Role Sketches.

## C  QA DETAILS

### C.1  AUTOMATICALLY QA GENERATION

The prompt shown in Figure 10 guides the Gemini in generating diverse and challenging Question-Answer (QA) pairs that form the foundation of SIV-Bench. Its design ensures these QAs effectively cover our core evaluation dimensions.

To create the multiple-choice options for each QA pair, we utilize a dedicated prompt for distractor generation, as detailed in Figure 11. This prompt instructs GPT to produce four unique distractor options based on the given question and its corresponding correct answer. Crucially, the prompt emphasizes that these distractors must be reasonable and clearly distinguishable from the correct answer, while also maintaining consistency with the correct answer in terms of sentence length, language style, and grammar. Furthermore, the prompt specifies a JSON output format for the list of distractors to ensure seamless programmatic integration. This structured approach to distractor generation is essential for constructing high-quality, fair multiple-choice questions that minimize biases arising from option formatting and robustly test model comprehension.

You will be given a video depicting human social relationships. Your task is to generate **question-answer (QA) pairs** to test the model's social reasoning ability. Each question must be clear and focused, addressing only one aspect at a time.

**Question Requirements**
- The questions should be challenging, requiring complex reasoning, an understanding of social norms, or real-world knowledge. Avoid surface-level observations.
- Generate **8-10 QA pairs**, distributed as follows:
- Descriptive: Focus on different aspects of the scene, but you don't need to cover all of them. You can choose what you find most important:
- Verbal characteristics (e.g., tone, pitch, speech style, etc.)
- Non-verbal characteristics (e.g., facial expressions, body language, etc.)
- Environmental features (e.g., location, setting, background details, etc.)
- Human interaction elements (e.g., emotion, feelings, attitude, reactions, etc.)
The question should be specific and unambiguous. Avoid questions like "describe the non-verbal characteristics of the video."
- Explanatory: Questions that explore the reasons behind observed behaviors.
- Predictive: A question that asks what is likely to happen next.
- Counterfactual: A question that explores hypothetical changes. For example, consider how the interaction might change if the individuals in the video had a stronger or weaker relationship, or if their relationship were of a different nature."

**Answer Requirements**
- Give a confident answer. Do not use words like "could" or "might." Provide only the essential information without additional explanations. Avoid making assumptions about relationships or identities in the QA. For example, DO NOT use descriptions like "the father and son in the video."
The output should be formatted in **JSON** as follows:
```json
{ "qa_pairs": [ { "category": "Descriptive", "question": "...", "answer": "..." }, { "category": "Explanatory", "question": "...", "answer": "..." },
{ "category": "Predictive", "question": "...", "answer": "..." }, { "category": "Counterfactual", "question": "...", "answer": "..." } ] }
```

Figure 10: The prompt used to guide Gemini for the initial generation of question-answer pairs.

Given the following question and answer, generate four distractors that are reasonable and distinguishable from the correct answer.
The sentence length, language style and grammer should be consistent with the answer.\n\n"
f"Question: {qa_pair.get("question", "none")}\n"
f"Answer: {qa_pair.get("answer", "none")}\n\n"
"Provide the output in the following JSON format: \n"
"{\"distractors\": [\"<distractor 1>\", \"<distractor 2>\", \"<distractor 3>\", \"<distractor 4>\"]}"

Figure 11: The prompt used to generate distractors for each QA pair.

## C.2 AUTOMATICALLY QA FILTERING

To ensure that the QA pairs in SIV-Bench genuinely require understanding of the video content, particularly for Subset 1, we employ a video-agnostic filtering step. Figure 12 presents the prompt used to instruct GPT for this purpose. The central directive of this prompt is for the model to select the most likely answer from the provided options using only the textual content of the question and the options themselves, without any access to the video. The prompt also enforces a strict output format, requiring only the letter of the chosen answer, and instructs the model to make a random choice if it deems the information insufficient (though the primary test is correct identification based on text alone). If a QA pair is answered correctly under this video-absent condition, it indicates that the question may be solvable through world knowledge, commonsense reasoning, or cues within the text of the QA itself, rather than necessitating specific information from the video. Such QAs are subsequently removed from the candidate pool to enhance the benchmark's focus on video-grounded social understanding.

To further refine the SIV-Bench dataset, particularly in selecting challenging items for Subset 1, we employ GPT-4o-mini to assign a difficulty score to each Question-Answer (QA) pair. Figure 13 presents the prompt designed for this task. It directs the model to evaluate QA difficulty on a 1 (Very Easy) to 5 (Very Hard) scale, assuming the QA pertains to a social interaction video. The prompt outlines key assessment criteria, including the subtlety of relevant cues, the complexity of reasoning involved, and the depth of social understanding needed to arrive at the correct answer. This automated difficulty scoring enables us to systematically identify and prioritize QAs that demand more sophisticated comprehension, thereby enhancing the overall rigor of our benchmark.

For the following questions, select the one you think is the most likely from the options provided to you. If you think the question doesn't contain the necessary information that can be answered, then just randomly choose one.
You are only allowed to output **the LETTER of the answer**. No other text or explanations are permitted.

Question: "{question}"
Options: "{options}"

Strictly output the answer in the following format:
answer: THE ANSWER LETTER

Figure 12: Prompt for the video-agnostic filtering of Question-Answer pairs.

Your task is to evaluate the difficulty of the provided Question-Answer (QA) pair. Assume the question is meant to be answered based on understanding a corresponding social interaction video. Assign a difficulty score on a scale of 1 (Very Easy) to 5 (Very Hard).

When assessing difficulty, consider these factors:
- **Subtlety of Cues:** Does answering correctly depend on noticing subtle visual details, nuanced language, or implicit contextual information, as opposed to obvious and explicit cues?
- **Reasoning Complexity:** Is the answer obtainable through simple observation, or does it require multi-step inference, understanding cause-and-effect, or integrating multiple pieces of information?
- **Depth of Social Understanding Required:** Does the QA tap into superficial aspects, or does it necessitate a deeper understanding of social norms, relationships, emotions, intentions, or complex social dynamics?
- **Specificity and Lack of Ambiguity:** Is the question well-posed and leading to a clear answer, or does its perceived difficulty stem from vagueness (which is not desired)? (Focus on inherent difficulty assuming a clear question).

**Difficulty Scale:**
- **1 (Very Easy):** Requires only direct observation of highly salient information; minimal to no inference.
- **2 (Easy):** Involves straightforward recall or identification of explicit cues; simple, single-step inference might be needed.
- **3 (Moderate):** Needs some level of inference, connecting different cues, or understanding common social situations/scripts.
- **4 (Hard):** Demands significant inference, interpretation of subtle or nuanced multimodal cues, multi-step reasoning, or a sophisticated understanding of social context.
- **5 (Very Hard):** Requires profound comprehension of complex and possibly implicit social dynamics, sophisticated multi-step reasoning, and potentially the integration of considerable world knowledge or nuanced social commonsense.

Question: "{question}"
Answer: "{answer}"

Strictly provide the output in the following JSON format:
{
  "difficulty_score": <integer_from_1_to_5>
}

Figure 13: Prompt used to score the difficulty of SIV-Bench QA pairs on a 1-to-5 scale, based on defined criteria such as cue subtlety, reasoning complexity, and depth of social understanding required.

To classify each QA pair in SIV-Bench, we use a GPT model guided by a structured prompt (see Figure 14). The prompt defines our three main assessment dimensions—SSU, SSR, and SDP—and their sub-tasks. The model analyzes each QA and outputs its corresponding sub-task, ensuring consistent and precise categorization across the benchmark.

We explicitly acknowledge that employing specific MLLMs (Gemini-2.0-Flash and GPT-4o-mini) for QA generation, distractor creation, and filtering may introduce inherent stylistic priors. While we mitigated this via human verification and option standardization, the resulting data distribution may still align more closely with the reasoning patterns of these generator models. This potential 'self-preference' bias should be considered when interpreting the relative performance of these specific model families.

## C.3  HUMAN ANNOTATION

**Annotator Information.** For the human annotation tasks integral to SIV-Bench, we recruit a team of 20 annotators. This cohort consists of 12 female and 8 male individuals, with an average age of 27 years. All annotators are well-educated and demonstrate proficiency in English, which is the

For the understanding and reasoning of a social interaction video, it needs to be examined from the following three aspects of ability:

**SSU** (Social Scene Understanding): Action Recognition, Facial Expression Recognition, Human Attribute Identification, Environment Perception;

**SSR** (Social State Reasoning): Relation Inference, Emotion Inference, Intent Inference, Attitude Inference;

**SDP** (Social Dynamics Prediction): Factual Prediction, Counterfactual Prediction;

Now you need to analyze and categorize a multiple-choice question, output which level of test it belongs to and what aspects it specifically tests.

Here is the content:

Question: "{question}"
Options: "{options}"

Your output should be in the following format:
SSR: Relation Inference.

No other text or explanations are permitted.

Figure 14: Prompt used to instruct LLM for the final classification of Question-Answer pairs into one of the 10 fine-grained sub-tasks under SSU, SSR and SDP.

primary language for instructions and annotations. To ensure fair compensation for their detailed work, annotators are remunerated at an hourly rate of $4 USD.

**Annotator Guidelines.** To maintain consistency and high quality in human annotation, particularly for the creation of new challenging Question-Answers (QAs) for Subset 2, annotators are provided with a comprehensive set of guidelines, as illustrated in Figure 15. These instructions detail the process for responding to existing multiple-choice questions and, crucially, guide annotators in formulating one new challenging question per video. For this QA creation task, annotators are directed to ensure their questions primarily test one of our three core assessment dimensions: Social Scene Understanding (SSU), Social State Reasoning (SSR), or Social Dynamics Prediction (SDP). The guidelines, including example questions, also emphasize that new QAs must be demanding, clearly worded, and require a deep understanding of the video content rather than superficial observation.

**Annotation Process.** All human annotation tasks for SIV-Bench, including the review of existing Question-Answers (QAs) and the generation of new QAs, are conducted using a custom-designed web interface. Figure 16 provides a representative example of this interface, which allows annotators to view the social interaction video, respond to provided multiple-choice questions, and submit their own newly authored questions and answers based on the video content and provided guidelines. This platform ensures a standardized environment for all human annotation contributions.

**Quality Control and Inter-Annotator Agreement.** To quantitatively address label quality, we computed the Inter-Annotator Agreement (IAA) using Fleiss' Kappa. For the initial set of approximately 27,000 QA pairs generated by our three LLM experts, the resulting Fleiss' Kappa was 0.52, indicating "moderate" agreement. This confirmed our expectation of ambiguity in the automated generation and validated the need for our strict inter-model consensus filter to create the high-quality Subset 1. For the more challenging Subset 2, where the LLMs disagreed, we measured the agreement between our human annotators, yielding a Fleiss' Kappa of 0.68. According to established standards (Landis & Koch, 1977), this value represents "substantial" agreement. Achieving this on the most difficult portion of our dataset confirms that humans can establish a reliable ground truth and justifies our methodology of retaining only those questions on which all assigned annotators unanimously agreed for the final benchmark.

## C.4  QA STATISTICS

To ensure the quality and balance of our SIV-Bench Question-Answer (QA) pairs, we conducted a statistical analysis of their structural properties, as summarized in Figure 17. This analysis examines several aspects: the average word count per multiple-choice option (Figure 17(a)) is relatively consistent across options A through E, minimizing length-based cues. The distribution of correct answers (Figure 17(b)) is fairly uniform across the five options, preventing positional bias. Furthermore, the overall question length (Figure 17(c)) peaks at around 11 words but shows a broad range, indicating variability in question complexity. Finally, an analysis of the first word in questions (Figure 17(d)) reveals a diverse set of interrogative types, led by 'what,' 'why,' and 'which,' reflecting a variety of reasoning challenges posed to the models.

## C.5  QUALITY AND DIVERSITY ANALYSIS

To further validate the quality of SIV-Bench, we conducted two analyses to ensure our benchmark encourages genuine comprehension over fitting to superficial patterns.

### C.5.1  ANALYSIS OF TEMPLATE PATTERN EXPLOITATION

SIV-Bench organizes tasks into SSU, SSR, and SDP to evaluate distinct dimensions of social cognition. To directly test whether models exploit surface-level cues, we analyzed the properties of questions that a representative model (Gemini-2.0-Flash) answered correctly versus incorrectly. As shown in Table 5, while there are statistically significant differences in length and word count, the small effect sizes (Cohen's d) indicate no meaningful structural separation between the two groups. Furthermore, Table 6 shows a high cosine similarity between the embeddings of correct and incorrect QA sets, alongside similar internal distributions (intra-similarity and variance). This close alignment suggests that model performance is unlikely to rely on superficial statistical or semantic patterns.

Table 5: Length and word count comparison between correctly and incorrectly answered questions.

| Metric | Correct (Mean ± SD) | Wrong (Mean ± SD) | t-stat | p-value | Cohen's d |
|---|---|---|---|---|---|
| Mean Length (chars) | 251.28 ± 93.88 | 228.96 ± 94.49 | 9.05 | 2.36e-19 | 0.2373 |
| Word Count | 42.13 ± 15.50 | 38.21 ± 15.78 | 9.54 | 2.63e-21 | 0.2516 |

Table 6: Semantic similarity analysis between correctly and incorrectly answered question sets.

| Metric | Value |
|---|---|
| Cosine Similarity (TF-IDF) | 0.9765 |
| Cosine Similarity (SentenceTransformer) | 0.9759 |
| Correct Intra-similarity | 0.1689 |
| Wrong Intra-similarity | 0.1711 |
| Correct Embedding Variance | 0.002164 |
| Wrong Embedding Variance | 0.002157 |

### C.5.2  LINGUISTIC DIVERSITY ANALYSIS

The majority of questions in SIV-Bench are generated by large language models or written by human annotators, rather than using rigid templates. To objectively measure our benchmark's linguistic diversity, we compared it against several prominent video QA benchmarks using two standard metrics: **Mean Semantic Distance** (average pairwise cosine distance of sentence embeddings) and **Vector Variance** (average variance across embedding dimensions). As shown in Table 7, SIV-Bench exhibits high semantic diversity, ranking among the top benchmarks. These results support that SIV-Bench's questions are varied and not limited to shallow templates, thereby promoting genuine semantic understanding.

Table 7: Semantic diversity comparison across video QA benchmarks.

| Benchmark | Mean Semantic Distance ↑ | Vector Variance ↑ |
|---|---|---|
| **SIV-Bench** | **0.8321** | **0.0022** |
| Video-Bench | 0.8811 | 0.0023 |
| Perception_Test | 0.7677 | 0.0020 |
| VideoVista | 0.7604 | 0.0020 |
| Social-IQ 2.0 | 0.7260 | 0.0019 |
| MVBench | 0.7124 | 0.0019 |
| EgoSchema | 0.5411 | 0.0014 |

## D EXPERIMENTS DETAILS

### D.1 SETTINGS DETAILS

This section provides further details on our experimental setup for evaluating MLLMs on SIV-Bench. Figure 18 displays the standardized prompt templates employed for model evaluations, with distinct versions tailored to models based on their input capabilities. For models that process sequences of images, the "PROMPT for frames input" (Figure 18, Top) informs the MLLM that it will receive a set of uniformly sampled frames from a video in chronological order. For models capable of direct video processing, the "PROMPT for videos input" (Figure 18, Bottom) is used. Both prompts clearly instruct the MLLM on its role, the task of answering multiple-choice questions based on the provided visual input, the expected JSON-like format for organizing answers (providing the exact text of the chosen option for each question), and a strict directive to avoid any extraneous text such as explanations or conversational remarks. This standardized, yet input-adaptive, prompting approach ensures consistency in task presentation across different model architectures.

For all evaluations, the specific inference parameters used for each model—such as temperature, top-p, or maximum new tokens—are adopted from their default configurations as provided within the VLMEvalKit (Duan et al., 2024) framework. This adherence to default settings aims to reflect the out-of-the-box capabilities of these models and ensure fair comparability. The experiments are conducted on two primary compute clusters. Cluster 1, utilized for evaluating the largest open-source models (Qwen2.5-VL-72B-Instruct and InternVL-78B), is equipped with an AMD EPYC 7642 48-Core Processor and 4x NVIDIA A100 GPUs. The total runtime for the reported experiments on this cluster is approximately 3 days. Cluster 2, used for the remaining models, consists of an Intel(R) Xeon(R) Platinum 8369B CPU @ 2.90GHz and 8x NVIDIA RTX 3090 GPUs. The cumulative runtime for experiments on this cluster is approximately 2 days. It should be noted that the overall research project, including preliminary testing on earlier dataset versions and exploratory experiments not included in the final results, involved a greater amount of compute time than the specific durations reported for the final benchmark evaluations.

### D.2 RESULTS DETAILS

**Fine-grained Analysis of Relation Inference (RI).** To provide granular insight into the critical Relation Inference sub-task and quantitatively verify the failure patterns discussed in Section 3.3, we further break down model performance across the 14 foundational social relationship types. Table 11 details the accuracy of each model per relation. This breakdown reveals significant variance; models perform robustly on visually distinct relations (e.g., *Couple*, *Team*) but struggle with nuanced dynamics like *Transactional* or *Caregiver-Recipient*.

Figure 19 further illustrates the aggregated confusion matrix for these predictions. The matrix exposes systematic misclassification tendencies rooted in social context. Specifically, we observe **Authority vs. Equality confusion**, where hierarchical roles like *Boss-Employee* and *Coach-Player* are notably misclassified as their egalitarian equivalents (*Colleagues* and *Teammates*). Additionally, **Intra-Group confusion** appears within categories; for instance, *Caregiver-Recipient* is often confused with *Parent-Child*, suggesting models detect the general affect of care but lack the precision to distinguish the specific relational setting.

Table 8: Comparative accuracy (%) of MLLMs on the 10 fine-grained sub-tasks in SIV-Bench (**Origin**), grouped by SSU, SSR, and SDP dimensions.

| Models | SSU | | | | SSR | | | | SDP | | Overall |
|---|---|---|---|---|---|---|---|---|---|---|---|
| | AR | FER | HAI | EP | II | EI | AI | RI | FP | CP | |
| mPLUG-Owl3 | 62.36 | 68.35 | 59.50 | 70.61 | 62.81 | 65.08 | 61.17 | 60.13 | 56.98 | 74.98 | 63.79 |
| LLaVA-OneVision | 59.55 | 65.60 | 71.90 | 59.34 | 70.36 | 55.79 | 56.74 | 61.18 | 70.06 | 58.68 | 63.73 |
| LLaVA-Video | 60.11 | 63.76 | 71.07 | 78.41 | 64.17 | 61.22 | 56.54 | 60.05 | 65.19 | 61.48 | 63.18 |
| Qwen2.5-VL-7B-Instruct | 58.71 | 64.22 | 70.25 | 81.26 | 68.15 | 68.21 | 59.96 | 56.58 | 64.99 | 63.21 | 65.01 |
| InternVL3-8B | 64.05 | 72.48 | 67.77 | 82.47 | 70.08 | 70.32 | 61.77 | 53.03 | 60.90 | 70.62 | 66.14 |
| Qwen2.5-VL-72B-Instruct | 86.52 | 85.78 | 80.99 | 84.01 | 71.72 | 75.81 | 67.81 | 66.99 | 74.76 | 77.04 | 74.25 |
| InternVL3-78B | 82.30 | 81.65 | 82.65 | 82.16 | 73.78 | 73.15 | 63.18 | 61.02 | 72.15 | 74.32 | 72.16 |
| o4-mini | 86.80 | 90.83 | 77.69 | 87.26 | 72.42 | 67.46 | 65.80 | 64.73 | 72.05 | 73.83 | 72.30 |
| GPT-4o | 88.48 | 84.40 | 80.99 | 88.67 | 70.45 | 78.02 | 66.20 | 64.65 | 74.41 | 75.31 | 73.16 |
| Gemini-2.0-Flash | 90.73 | 83.03 | 85.12 | 85.21 | 74.11 | 67.03 | 67.20 | 64.41 | 69.87 | 77.04 | 72.15 |
| Gemini-2.5-Flash | 91.29 | 85.32 | 87.60 | 88.20 | 74.48 | 67.04 | 67.61 | 64.97 | 71.86 | 78.03 | 73.67 |
| Gemini-2.5-Pro | 93.54 | 91.74 | 89.26 | 88.08 | 75.14 | 76.94 | 69.62 | 67.80 | 72.01 | 79.18 | 76.03 |

Table 9: Comparative accuracy (%) of MLLMs on the 10 fine-grained sub-tasks in SIV-Bench (**+sub**), grouped by SSU, SSR, and SDP dimensions.

| Models | SSU | | | | SSR | | | | SDP | | Overall |
|---|---|---|---|---|---|---|---|---|---|---|---|
| | AR | FER | HAI | EP | II | EI | AI | RI | FP | CP | |
| mPLUG-Owl3 | 62.60 | 68.55 | 59.69 | 70.83 | 63.03 | 65.30 | 61.36 | 60.33 | 57.17 | 75.22 | 64.01 |
| LLaVA-OneVision | 60.22 | 66.25 | 72.59 | 60.01 | 71.03 | 56.47 | 57.41 | 61.88 | 70.71 | 59.37 | 64.40 |
| LLaVA-Video | 60.65 | 64.30 | 71.61 | 78.98 | 64.75 | 61.75 | 57.12 | 60.63 | 65.76 | 62.07 | 63.75 |
| Qwen2.5-VL-7B-Instruct | 58.81 | 64.34 | 70.35 | 81.34 | 68.29 | 68.34 | 60.08 | 56.68 | 65.10 | 63.35 | 65.13 |
| InternVL3-8B | 64.21 | 72.60 | 67.90 | 82.65 | 70.27 | 70.44 | 61.89 | 53.17 | 61.01 | 70.75 | 66.28 |
| Qwen2.5-VL-72B-Instruct | 87.02 | 86.31 | 81.52 | 84.53 | 72.23 | 76.34 | 68.32 | 67.51 | 75.28 | 77.55 | 74.77 |
| InternVL3-78B | 82.87 | 82.21 | 83.20 | 82.69 | 74.31 | 73.66 | 63.73 | 61.57 | 72.69 | 74.90 | 72.70 |
| o4-mini | 86.91 | 90.94 | 77.81 | 87.39 | 72.54 | 67.62 | 65.90 | 64.88 | 72.18 | 74.00 | 72.43 |
| GPT-4o | 89.60 | 85.51 | 82.12 | 89.83 | 71.53 | 79.17 | 67.38 | 65.78 | 75.53 | 76.45 | 74.29 |
| Gemini-2.0-Flash | 91.83 | 84.12 | 86.22 | 86.34 | 75.22 | 68.15 | 68.31 | 65.50 | 70.98 | 78.14 | 73.27 |
| Gemini-2.5-Flash | 91.42 | 85.44 | 87.72 | 88.34 | 74.63 | 67.19 | 67.73 | 65.09 | 72.00 | 78.19 | 73.82 |
| Gemini-2.5-Pro | 94.02 | 92.25 | 89.77 | 88.52 | 75.61 | 77.43 | 70.09 | 68.25 | 72.43 | 79.66 | 76.50 |

## D.3 FAILURE CASES

This section presents several illustrative failure cases. We focus on examples from Gemini-2.0-Flash, a strong closed-source model, to highlight that even advanced MLLMs can fail on nuanced aspects of social perception, prediction, and reasoning. These examples are categorized by our primary assessment dimensions: SSU, SSR, and SDP, and are intended to offer concrete instances for future research and model development.

Figure 20 illustrates instances where Gemini-2.0-Flash fails on SSU tasks, which require accurate perception of explicit visual elements. **(a) In Action Recognition**, the model incorrectly identifies the man's gesture as "He crosses his arms tightly" instead of the correct "He raises one eyebrow slightly", missing a subtle but distinct facial action. **(b) For Environment Perception**, when asked about the weather, the model failed to capture the details of the characters in the scene wearing thick scarves and down jackets to infer that the correct answer was "cold", but instead wrongly chose "wet". **(c) In Facial Expression Recognition**, the model describes the expression as "A mischievous smile" rather than the correct "A stoic glare", misinterpreting the nuanced facial expression display. **(d) For Human Attribute Identification**, concerning the child's clothing, the model selects "A dress" instead of the correct "A set of pajamas", failing to correctly identify common apparel.

Figure 21 presents failure cases of Gemini-2.0-Flash on SSR tasks, which involve inferring unobservable mental states and relationships. **(e) In Intent Inference**, when a woman says "do you understand?" to a boy who bullies her son in an angry tone, it is to teach him a lesson and warn him not to bully her son again, not for "discourage any defiance", because in fact no child much younger than her can form defiance against her. **(f) For Emotion Inference**, this employee is happy instead of scared after leaving because he successfully deceives the boss into giving him a vacation. **(g) In Attitude Inference**, the coworker is dissatisfied and disappointed with the cashier's nervousness,

Table 10: Comparative accuracy (%) of MLLMs on the 10 fine-grained sub-tasks in SIV-Bench (**-sub**), grouped by SSU, SSR, and SDP dimensions.

| Models | SSU | | | | SSR | | | | SDP | | Overall |
|---|---|---|---|---|---|---|---|---|---|---|---|
| | AR | FER | HAI | EP | II | EI | AI | RI | FP | CP | |
| mPLUG-Owl3 | 61.77 | 67.76 | 58.92 | 70.03 | 62.26 | 64.52 | 60.60 | 59.58 | 56.41 | 74.42 | 63.22 |
| LLaVA-OneVision | 57.98 | 64.00 | 70.30 | 57.73 | 68.77 | 54.20 | 55.15 | 59.58 | 68.45 | 57.10 | 62.14 |
| LLaVA-Video | 58.59 | 62.24 | 69.51 | 76.89 | 62.64 | 59.68 | 55.00 | 58.52 | 63.71 | 59.93 | 61.66 |
| Qwen2.5-VL-7B-Instruct | 57.23 | 62.72 | 68.78 | 79.79 | 66.62 | 66.71 | 58.48 | 55.10 | 63.53 | 61.76 | 63.53 |
| InternVL3-8B | 63.26 | 71.69 | 66.99 | 81.66 | 69.30 | 69.57 | 60.96 | 52.22 | 60.12 | 69.86 | 65.35 |
| Qwen2.5-VL-72B-Instruct | 85.83 | 85.08 | 80.28 | 83.31 | 71.02 | 75.11 | 67.09 | 66.32 | 74.03 | 76.29 | 73.54 |
| InternVL3-78B | 81.70 | 81.06 | 82.09 | 81.56 | 73.22 | 72.56 | 62.57 | 60.39 | 71.55 | 73.71 | 71.56 |
| o4-mini | 86.10 | 90.11 | 76.97 | 86.57 | 71.73 | 66.75 | 65.09 | 63.99 | 71.35 | 73.12 | 71.59 |
| GPT-4o | 88.43 | 84.37 | 80.94 | 88.64 | 70.39 | 77.96 | 66.20 | 64.60 | 74.38 | 75.22 | 73.12 |
| Gemini-2.0-Flash | 90.24 | 82.50 | 84.63 | 84.73 | 73.55 | 66.55 | 66.73 | 63.89 | 69.35 | 76.52 | 71.65 |
| Gemini-2.5-Flash | 90.17 | 84.20 | 86.45 | 87.07 | 73.34 | 65.87 | 66.46 | 63.85 | 70.71 | 76.90 | 72.53 |
| Gemini-2.5-Pro | 92.69 | 90.84 | 88.36 | 87.15 | 74.24 | 76.08 | 68.72 | 66.90 | 71.10 | 78.28 | 75.14 |

Table 11: Detailed accuracy breakdown of MLLMs on the Relation Inference (RI) task across 14 specific social relationship types. The "RI" column denotes the average accuracy.

| Model | Avg RI | Boss | Class | Care | Coach | Coll | Coup | Frnd | Grand | Par | Serv | Sib | Team | Teach | Trans |
|---|---|---|---|---|---|---|---|---|---|---|---|---|---|---|---|
| mPLUG-Owl3 | 60.13 | 61.74 | 51.39 | 77.52 | 74.52 | 44.02 | 72.52 | 42.52 | 49.55 | 42.52 | 55.68 | 42.52 | 49.60 | 42.52 | 42.52 |
| LLaVA-OneVision | 61.18 | 29.63 | 63.24 | 59.63 | 29.63 | 54.49 | 64.39 | 61.66 | 80.80 | 69.02 | 90.46 | 29.63 | 90.88 | 60.79 | 32.71 |
| LLaVA-Video | 60.05 | 28.46 | 78.56 | 53.46 | 44.46 | 67.34 | 77.51 | 67.92 | 78.73 | 39.37 | 69.08 | 37.03 | 73.04 | 36.37 | 54.61 |
| Qwen2.5-VL-7B-Instruct | 56.58 | 32.64 | 32.64 | 38.73 | 32.64 | 39.25 | 55.50 | 81.51 | 68.89 | 82.94 | 36.11 | 54.20 | 79.31 | 73.11 | 32.64 |
| InternVL3-8B | 53.03 | 21.89 | 59.62 | 45.37 | 36.51 | 56.02 | 32.84 | 54.87 | 35.22 | 49.16 | 77.40 | 48.64 | 78.97 | 78.63 | 59.47 |
| Qwen2.5-VL-72B-Instruct | 66.99 | 63.08 | 95.00 | 67.09 | 52.37 | 72.28 | 46.70 | 73.77 | 75.66 | 58.34 | 76.17 | 43.88 | 68.72 | 36.45 | 81.98 |
| InternVL3-78B | 61.02 | 66.91 | 57.39 | 69.18 | 82.31 | 55.95 | 55.95 | 83.15 | 70.97 | 52.42 | 67.60 | 52.51 | 52.48 | 63.09 | 30.76 |
| gpt-4o-mini | 64.73 | 60.47 | 86.19 | 36.65 | 69.65 | 63.33 | 54.03 | 66.51 | 85.43 | 70.17 | 47.71 | 59.34 | 54.83 | 68.11 | 77.03 |
| GPT-4o | 64.65 | 49.24 | 95.00 | 68.95 | 61.95 | 53.20 | 63.95 | 63.95 | 33.95 | 60.01 | 89.08 | 49.01 | 72.28 | 44.18 | 41.64 |
| Gemini-2.0-Flash | 64.41 | 64.61 | 83.30 | 21.86 | 45.86 | 63.54 | 37.57 | 72.81 | 79.64 | 75.19 | 43.43 | 69.39 | 81.03 | 78.60 | 77.24 |
| Gemini-2.5-Flash | 64.97 | 80.68 | 64.21 | 28.86 | 72.29 | 55.53 | 66.96 | 63.75 | 84.67 | 77.47 | 62.59 | 57.43 | 29.56 | 78.07 | 86.23 |
| Gemini-2.5-Pro | 67.80 | 67.95 | 75.51 | 29.85 | 77.28 | 70.44 | 79.85 | 72.98 | 91.37 | 82.66 | 63.62 | 61.28 | 36.28 | 76.28 | 58.57 |

panic and even physical reactions when seeing female customers. This could also be seen from his subsequent warning to the cashier not to do so anymore. **(h) For Relation Inference**, we present case studies on the failure patterns of the four common models listed in the main text in this task.

Figure 22 highlights errors made by Gemini-2.0-Flash in SDP tasks, which require predicting future events or reasoning about hypothetical scenarios. **(i) In Factual Prediction**, when asked if the person in the black shirt would be satisfied with the workers' work, since the two of them have already reached an agreement with smiles at the end of the video, it could be inferred that the answer was "yes", but the model chooses another answer. **(j) For Counterfactual Prediction**, the video shows the dance interaction between a mother and her son. The question raised is what would happen if one of them had more dance experience. This can be inferred from the positive and relaxed interaction between the two in the video. The most likely answer is "The more experienced dancer leads and adapts movements". For example, a son leads his mother to learn dancing happily, rather than "The less experienced dancer hesitates and struggles to keep up". They have a good relationship, and the probability of negative performance like "hesitates" and "struggles" is lower.

### D.4 ANALYSIS OF CHAIN-OF-THOUGHT PROMPTING

To investigate the impact of explicit reasoning on model performance, we conducted a preliminary experiment using a Chain-of-Thought (CoT) prompting strategy. We prepended the instruction *"Let's think step by step. First, output your reasoning process, and then output the final answer."* to our standard evaluation prompt. The overall accuracy on the `origin` videos, with and without CoT, is presented in Table 12.

The results indicate that applying a generic CoT prompt did not yield significant performance improvements for most models. For several smaller open-source models (e.g., mPLUG-Owl3, LLaVA-Video), it resulted in a notable performance decrease. We observed that this is often because these models struggle to consistently adhere to the more complex two-stage output format (i.e.,

providing reasoning before the final answer in the required format), leading to failures in our answer parsing logic.

The primary goal of SIV-Bench is to establish a fair, consistent, and reproducible evaluation of baseline model capabilities. Our standardized prompting strategy, which aligns with widely used toolkits like VLMEvalKit, ensures this fairness. While techniques like CoT are powerful for eliciting maximum performance from certain capable models (e.g., Gemini-2.5-Pro), introducing them as a default can create a confounding variable. Such a setup might shift the evaluation from testing inherent social reasoning to testing complex instruction-following abilities. Therefore, our main experiments use a direct-answering prompt to maintain a level playing field. Nonetheless, these findings suggest that developing more specialized reasoning methods tailored to social intelligence is a valuable direction for future work that builds upon this benchmark.

Table 12: Comparison of Overall Accuracy (%) on the `origin` videos with and without Chain-of-Thought (CoT) prompting.

| Model | Acc (Origin) | Acc (CoT) |
|---|---|---|
| mPLUG-Owl3 | 63.79% | 61.52% |
| LLaVA-OneVision | 63.73% | 62.98% |
| LLaVA-Video | 63.18% | 62.22% |
| Qwen2.5-VL-7B-Instruct | 65.01% | 65.75% |
| InternVL-8B | 66.14% | 65.31% |
| Qwen2.5-VL-72B-Instruct | 74.25% | 74.58% |
| InternVL-78B | 72.16% | 71.95% |
| o4-mini | 72.30% | 72.41% |
| GPT-4o | 73.76% | 73.65% |
| Gemini-2.0-Flash | 72.75% | 72.88% |
| Gemini-2.5-Flash | 73.67% | 73.51% |
| Gemini-2.5-Pro | 76.03% | 76.15% |

## D.5 STATISTICAL SIGNIFICANCE ANALYSIS

To validate the reliability of our comparative claims, we conducted McNemar's tests on the full dataset ($N = 8,728$). This paired non-parametric test is appropriate for comparing the performance of two classifiers on the same dataset.

**Model Ranking.** We verified the leadership of our SOTA model. The performance difference between the top-performing **Gemini-2.5-Pro** (76.03%) and the second-best model **Qwen2.5-VL-72B** (74.25%) is highly statistically significant ($p < 0.001$), confirming the robustness of the leaderboard rankings.

**Task Difficulty.** We confirmed that the performance stratifications across our three core dimensions are not due to chance. For Gemini-2.5-Pro, the performance gaps between **SSU** (90.67%) and **SDP** (75.28%), as well as between **SDP** and **SSR** (70.20%), are all highly statistically significant ($p < 0.001$).

**Audio Impact.** We validated the contribution of non-textual audio cues (Table 3). The performance degradation observed when removing audio from original videos (e.g., $73.67\% \rightarrow 71.95\%$ for Gemini-2.5-Flash) is statistically significant ($p < 0.01$).

**Subtitle Influence.** We performed significance testing on the subtitle conditions (Table 4). Our analysis reveals that the minor *Overall* improvement from adding subtitles ('+sub') is not statistically significant ($p = 0.12$). However, the impact is task-dependent: the negative impact of removing text ('-sub') is statistically significant for the **SSR** task ($p < 0.05$), and the benefit of added subtitles ('+sub') is significant for the **SDP** task ($p < 0.05$).

# E   IN-DEPTH ANALYSIS ON SIV-BENCH-HARD

To rigorously evaluate the upper limits of current MLLMs, establish a robust human baseline, and probe the reasoning processes beyond simple accuracy, we curated and analyzed a challenging subset of our dataset, termed **SIV-Bench-Hard**. This section details the setup, human performance, and a multi-dimensional analysis of model reasoning quality on this subset.

## E.1   EXPERIMENTAL SETUP

**Dataset Curation.** We selected a subset of 200 questions from the human-generated portion of SIV-Bench. These questions were specifically chosen for their complexity and reliance on deep social understanding, filtering out items that could be solved via superficial visual cues.

**Task Definition.** Unlike the standard multiple-choice evaluation, this study required both human annotators and MLLMs to provide: (1) the selected answer option, and (2) a free-text *reasoning explanation* justifying their choice. This allows for a deeper examination of the cognitive process.

**Participants.** We recruited 3 independent human annotators to perform this task to establish a human baseline. We evaluated a suite of state-of-the-art MLLMs, including Gemini-3-Pro, GPT-5.1, Gemini-2.5-Pro, Gemini-2.5-Flash, Qwen2.5-VL-7B, and GPT-4o-mini.

## E.2   HUMAN-AI PERFORMANCE GAP

Table 13 compares the accuracy of human annotators against MLLMs on SIV-Bench-Hard.

Table 13: Accuracy comparison between Human Annotators and MLLMs on the SIV-Bench-Hard subset. The results highlight a significant performance gap.

| Subject | Accuracy |
|---|---|
| Human Annotator 1 | 67.00% |
| Human Annotator 2 | 65.50% |
| Human Annotator 3 | 70.50% |
| **Human (Average)** | **67.67%** |
| Gemini-3-Pro | 45.50% |
| GPT-5.1 | 39.00% |
| Gemini-2.5-Pro | 37.00% |
| Gemini-2.5-Flash | 32.32% |
| GPT-4o-mini | 29.00% |
| Qwen2.5-VL-7B-Instruct | 24.50% |

**Analysis.** As shown in Table 13, we established a human baseline of approximately 67.7%. In sharp contrast, the performance of SOTA MLLMs collapses on this challenging subset. The best-performing model, GPT-5.1, achieved only 39.00%, revealing a massive $\sim$29% performance gap compared to humans. This finding confirms that SIV-Bench is far from saturated and remains a significant challenge for even the most advanced models.

## E.3   EVALUATION OF REASONING QUALITY

To go beyond accuracy, we assessed the quality of the generated free-text explanations. We employed GPT-4o-mini as an LLM-Judge. For each sample, the judge was provided with the video context, the question, the model's answer, and the explanations from the three human annotators as a "gold standard" reference. The judge scored the model's reasoning on a scale of 1-5 across five dimensions, based on the core assumption that human reasoning processes are the standard for social intelligence.

**Gap and Skew in Reasoning.** Table 14 reveals a critical divergence in model capabilities. MLLMs score highly on structural metrics such as *Logical Coherence* ($\sim$4.65) and *Conciseness* ($\sim$4.89), indicating proficiency in generating well-formed text. However, their scores for cognitive metrics—specifically *Alignment with Human Reasoning* ($\sim$3.3) and *Depth of Analysis* ($\sim$3.3)—are

Table 14: Reasoning Quality scores (1-5) for MLLMs on SIV-Bench-Hard. Scores are evaluated by an LLM-Judge using human reasoning traces as the reference.

| Model | Relevance | Alignment w/ Human | Logical Coherence | Depth of Analysis | Conciseness | Overall Score |
|---|---|---|---|---|---|---|
| Gemini-3-Pro | **4.66** | **3.30** | **4.67** | **3.49** | 4.87 | **4.10** |
| GPT-5.1 | 4.58 | 3.29 | 4.65 | 3.26 | 4.88 | 4.00 |
| Gemini-2.5-Pro | 4.57 | 3.26 | 4.65 | 3.41 | 4.89 | 4.05 |
| Gemini-2.5-Flash | 4.48 | 3.17 | 4.55 | 3.22 | 4.87 | 3.95 |
| GPT-4o-mini | 4.45 | 3.20 | 4.56 | 3.12 | **4.91** | 3.90 |
| Qwen2.5-VL-7B-Instruct | 4.00 | 2.89 | 4.21 | 3.05 | 4.45 | 3.63 |

significantly lower. This demonstrates that while MLLMs can generate logically structured explanations, they struggle to replicate the analytical depth and specific cognitive patterns characteristic of human social reasoning.

### E.4 QUANTITATIVE ANALYSIS OF REASONING EMBEDDINGS

To quantify the divergence observed in the qualitative scores, we performed an embedding-based analysis. We encoded all reasoning texts (both human and model) using the `paraphrase-multilingual-MiniLM-L12-v2` model.

**Distinguishability.** We trained a Random Forest classifier to distinguish between human and model explanations based on their embeddings. The classifier achieved an accuracy of **92.17%** (compared to a 50% random baseline). This high classification accuracy indicates that the latent semantic features of model reasoning are fundamentally distinct from those of humans.

**Statistical Significance.** We further validated this difference using Mann-Whitney U tests across the embedding dimensions. The tests confirmed that the distributions of human and model embeddings are significantly different ($p < 0.05$) on 8 out of 10 principal dimensions. Collectively, these results provide quantitative evidence that current MLLMs, despite their linguistic fluency, employ reasoning processes that are statistically distinguishable from human social cognition.

The SIV-Bench-Hard subset, along with the human reasoning annotations and analysis code, will be released to facilitate future research into bridging this gap.

### E.5 CORRELATION BETWEEN ANSWER CORRECTNESS AND REASONING FIDELITY

To rigorously verify that model performance is grounded in genuine social reasoning rather than superficial visual shortcuts (e.g., background cues or object co-occurrence), we analyze the relationship between the correctness of a model's answer and the semantic quality of its reasoning trace. We hypothesize that if models were merely "guessing" via shortcuts, their generated explanations would lack alignment with human cognitive processes even when they fortuitously select the correct option.

We perform an embedding-based analysis on the *SIV-Bench-Hard* subset using the `paraphrase-multilingual-MiniLM-L12-v2` model. For each of the six evaluated models, we calculate the cosine similarity between the model's generated reasoning and the human expert ground truth. The results are stratified based on whether the model answers the multiple-choice question correctly or incorrectly.

As illustrated in Figure 23 and summarized in Table 15, we observe a statistically significant positive gap in similarity scores across 5 out of the 6 models. Notably, GPT-5.1 demonstrates the strongest effect, with a similarity gap of 0.066 ($p < 0.001$).

These findings provide empirical evidence that correctness in SIV-Bench is strongly correlated with human-like social reasoning. The significant degradation in reasoning alignment during failure cases suggests that models do not rely on "guessing" via superficial cues; rather, successful performance necessitates a cognitive process that mirrors human social understanding.

Table 15: Quantitative comparison of reasoning similarity to human ground truth between correct and incorrect responses. The **Gap** ($\Delta$) represents the increase in similarity when the model answers correctly. Significance is calculated using the Mann-Whitney U Test.

| Model | Similarity (Correct) | Similarity (Incorrect) | Gap ($\Delta$) | Significance ($p$-value) |
|---|---|---|---|---|
| GPT-5.1 | 0.572 | 0.506 | +0.066 | $< \mathbf{0.001}$ (***) |
| Qwen2.5-VL-7B-Instruct | 0.535 | 0.474 | +0.061 | $< \mathbf{0.01}$ (**) |
| GPT-4o-mini | 0.571 | 0.513 | +0.058 | $< \mathbf{0.01}$ (**) |
| Gemini-2.5-Pro | 0.541 | 0.498 | +0.043 | $< \mathbf{0.05}$ (*) |
| Gemini-2.5-Flash | 0.537 | 0.504 | +0.033 | $< \mathbf{0.05}$ (*) |
| Gemini-3-Pro | 0.527 | 0.494 | +0.033 | 0.067 (n.s.) |

## F BROADER IMPACT

SIV-Bench is designed to foster positive advancements in artificial social intelligence, potentially leading to more empathetic, context-aware, and collaborative AI systems for beneficial applications such as assistive technologies, improved human-AI teaming, and richer content understanding. However, enhancing AI's grasp of social dynamics also presents risks. These capabilities could be misused for sophisticated manipulation, disinformation, or invasive surveillance, and unaddressed biases in data could be amplified, leading to inequitable outcomes. We offer SIV-Bench as a research tool to transparently assess MLLM capabilities and limitations in the social domain, thereby encouraging the community to proactively consider these ethical challenges and develop robust safeguards alongside continued innovation in social AI.

# Multimodal Social Interaction Understanding and Reasoning Test

## Task Introduction

This task evaluates your ability to understand and reason about various aspects of social interactions depicted in videos, including the observable scene, the participants' potential mental states, and how the interaction unfolds or might change.

## Instructions for Answering and Creating Questions

Each video is approximately **40 seconds** long on average. After watching each video, you will answer **1-3 multiple-choice questions**. Please select the answer you believe is most correct based on the video content.

Additionally, you will be asked to create **one challenging new question** about the video. Your question should encourage deep thinking about the social interaction. Please aim for your question to primarily test one of the following broad areas of understanding:

**1. Social Scene Understanding (SSU):** Questions focusing on the observable aspects of the interaction.
*(e.g., "What are people doing or saying explicitly? What are their visible expressions or key attributes? What does the environment tell us about the context?")*

**2. Social State Reasoning (SSR):** Questions requiring reasoning about the unobservable mental states of the individuals or the social relationship between them.
*(e.g., "What might a person be feeling, intending, or believing? What is their attitude? What is the nature of their relationship?")*

**3. Social Dynamic Prediction (SDP):** Questions about how the interaction evolves, predicting future events, or considering hypothetical changes to the scenario.
*(e.g., "What is likely to happen next as a result of this interaction? If a key element were different, how might the outcome change?")*

Ensure your newly created question is challenging, clearly worded, and requires careful consideration of the video content, not just superficial observation.

## Example Questions You Might Be Asked or Could Create:

- What clues in the setting or attire suggest this might be a formal event? (Tests SSU)
- Why do the two individuals exhibit contrasting emotional expressions after the announcement? (Tests SSR)
- If the person in blue had not intervened, what would likely have been the immediate consequence for the group? (Tests SDP)

**Name:**

Start Quiz

Figure 15: Screenshot of the guidelines provided to human annotators, detailing the tasks of answering multiple-choice questions and creating new challenging questions about Social Scene Understanding (SSU), Social State Reasoning (SSR), and Social Dynamics Prediction (SDP).

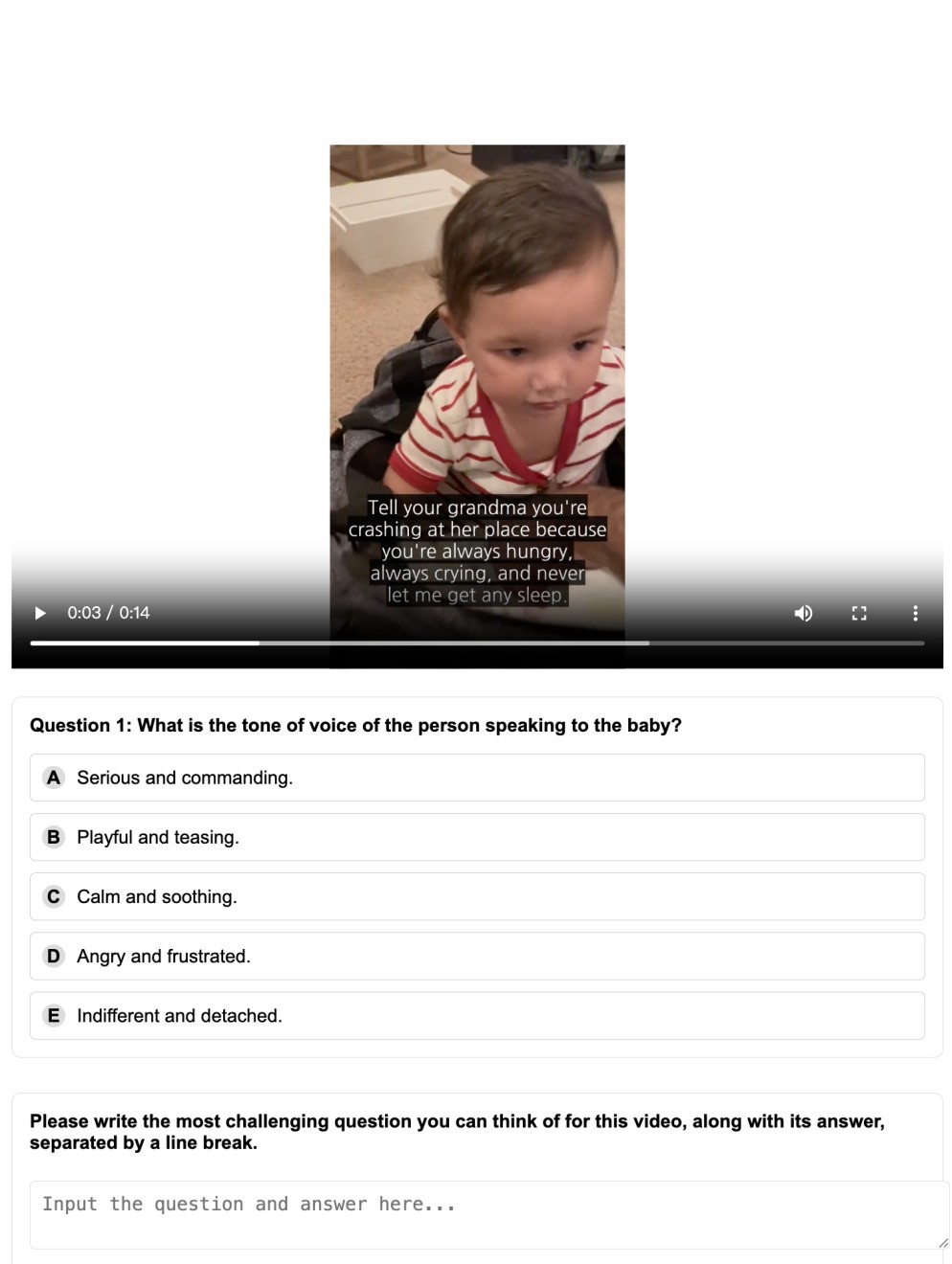

Figure 16: Example of the web-based interface used by human annotators for watching videos, answering provided multiple-choice questions, and authoring new Question-Answer pairs for SIV-Bench.

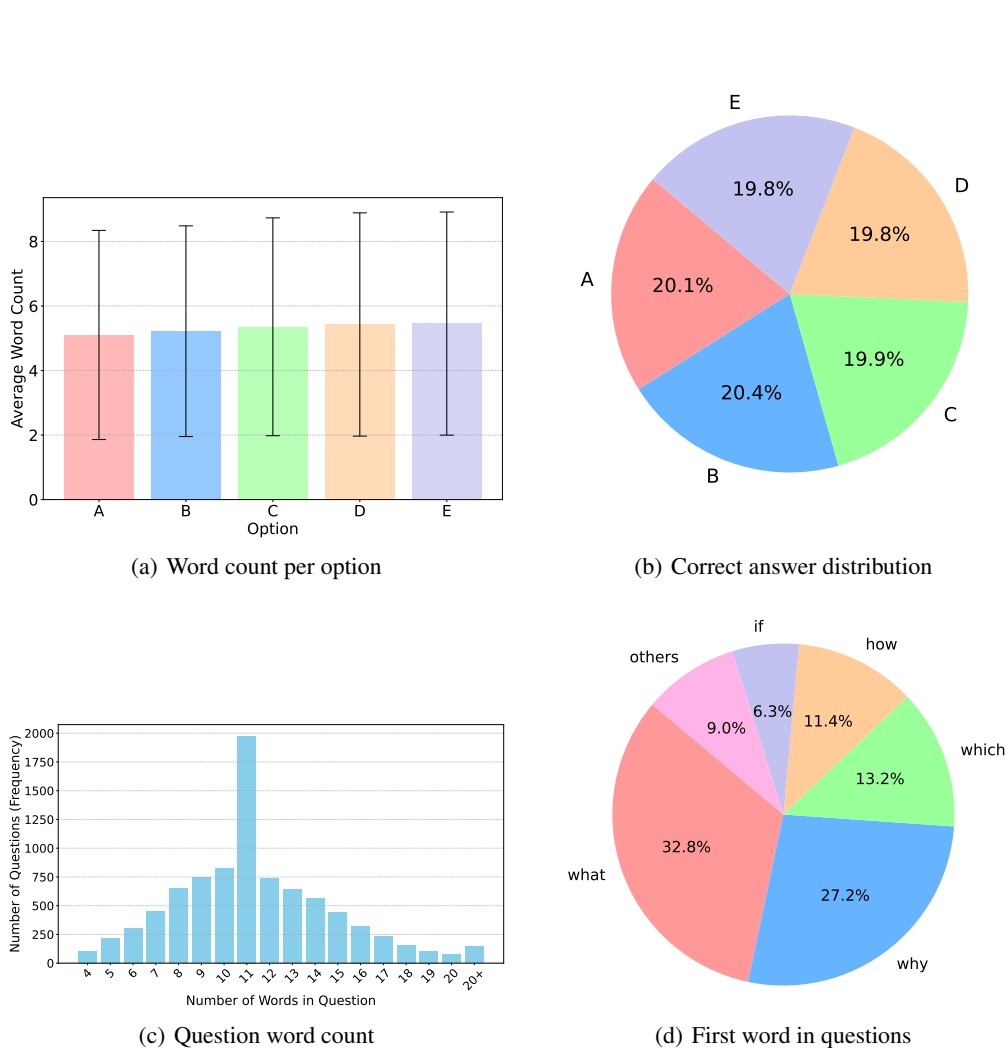

(a) Word count per option

(b) Correct answer distribution

(c) Question word count

(d) First word in questions

Figure 17: Statistical analysis of SIV-Bench Question-Answer (QA) pairs. (a) average word count consistency across answer options, (b) distribution of correct answers among options, (c) overall question length distribution by word count, and (d) frequency of common first words in questions.

**PROMPT for frames input**

You are an AI assistant responsible for answering questions about videos.

You will be provided with {} separate frames uniformly sampled from a video, \
the frames are provided in chronological order of the video.
Please analyze these images and provide the answers to the \
following multiple-choice questions about the video content.
If multiple questions are provided (with indices Q1, Q2, Q3, ...), \
you should organize your answers in the following json format:
1. [Exact text of the chosen option for question 1],
2. [Exact text of the chosen option for question 2],
...

Do NOT add any explanations, introductions, or concluding remarks.

**PROMPT for videos input**

You are evaluating a video based on the multiple-choice questions provided below.,
For each numbered question, select the best answer from the options listed.,
Your response MUST strictly follow this format:,
1. [Exact text of the chosen option for question 1],
2. [Exact text of the chosen option for question 2],
... and so on for all questions.

Do NOT add any explanations, introductions, or concluding remarks.,
--- QUESTIONS ---
…

Figure 18: Standardized prompt templates used for evaluating MLLMs on SIV-Bench. Separate prompts are shown for models that process (Top) uniformly sampled frames and (Bottom) direct video input.

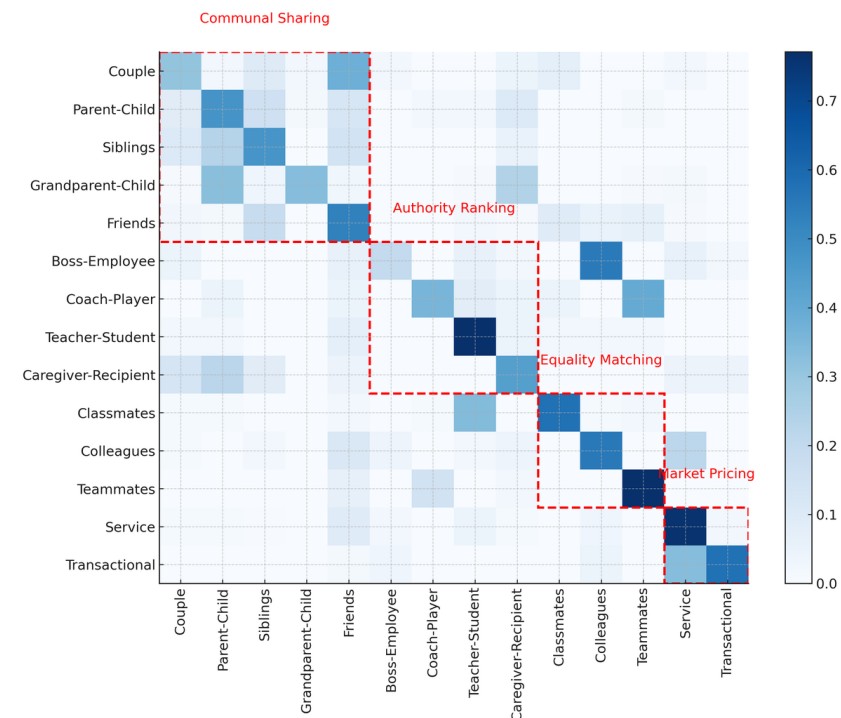

Figure 19: Aggregated Confusion Matrix for the Relation Inference (RI) task. Red dashed lines delineate the four foundational relational models. Off-diagonal clusters indicate systematic misclassifications, such as the confusion between Authority Ranking and Equality Matching relations.

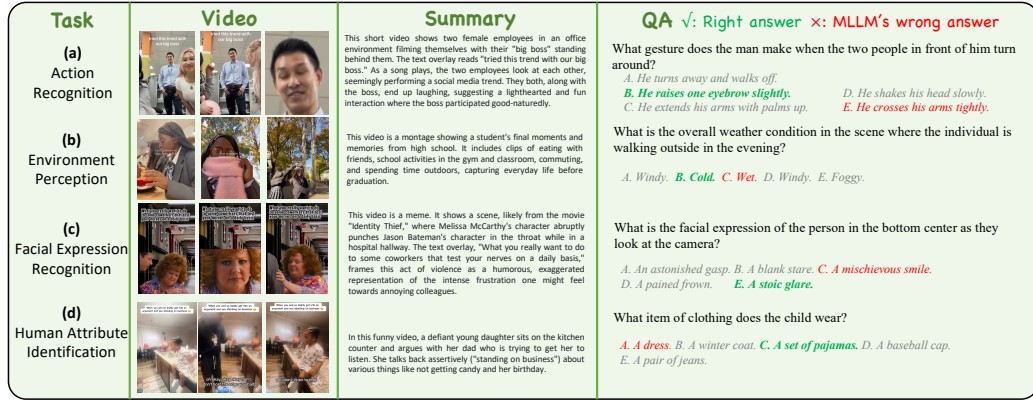

Figure 20: Examples of failure cases in Social Scene Understanding (SSU) tasks, including errors in Action Recognition, Environment Perception, Facial Expression Recognition, and Human Attribute Identification.

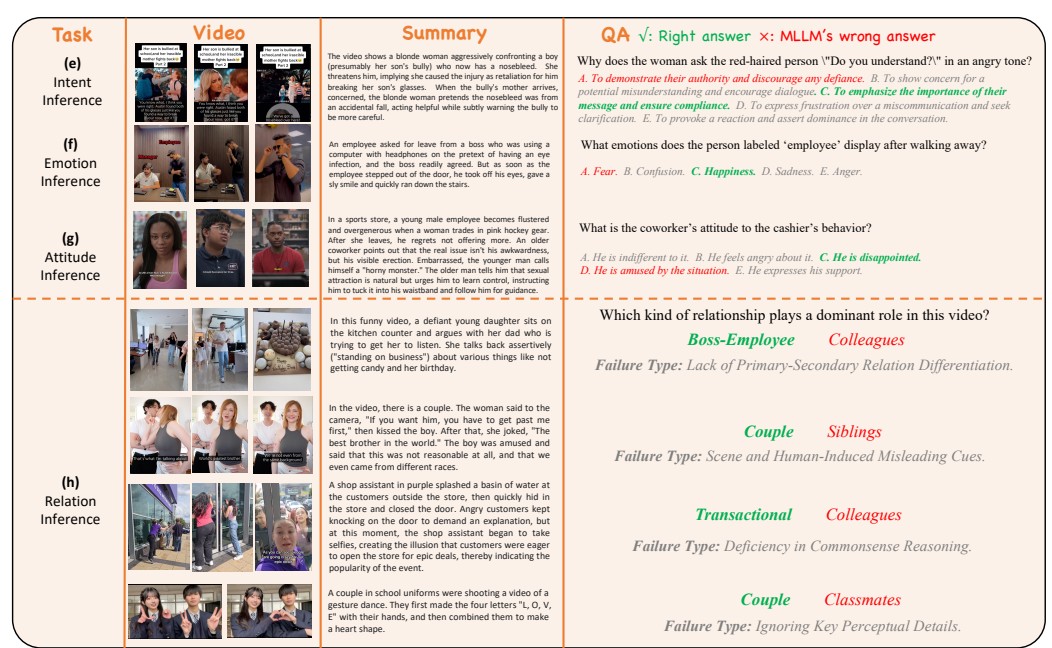

Figure 21: Examples of failure cases in Social State Reasoning (SSR) tasks, highlighting difficulties in Intent Inference, Emotion Inference, Attitude Inference, and Relation Inference.

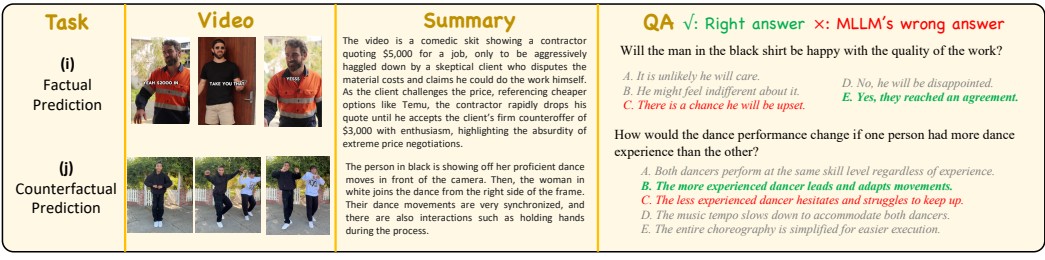

Figure 22: Examples of failure cases in Social Dynamics Prediction (SDP) tasks, covering both Factual Prediction and Counterfactual Prediction.

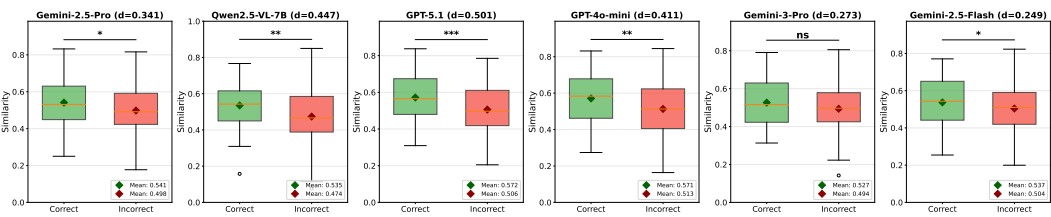

Figure 23: Box plots illustrating the distribution of semantic similarity scores between model reasoning and human ground truth, stratified by answer correctness. We observe a consistent trend where reasoning traces for correct answers exhibit significantly higher alignment with human social cognition compared to incorrect answers. Statistical significance is denoted by * ($p < 0.05$), ** ($p < 0.01$), and *** ($p < 0.001$).

