# OpenReview forum: "SIV-Bench: A Video Benchmark for Social Interaction Understanding and Reasoning"
_ICLR.cc/2026/Conference — ICLR 2026 Conference Withdrawn Submission_

### Official Review · Reviewer_A6Cm · 2025-10-29

**Soundness:** 3
**Presentation:** 3
**Contribution:** 2
**Rating:** 2
**Confidence:** 4

**Summary:**

This paper introduces SIV-Bench, a video benchmark for evaluating MLLMs on social interaction understanding across three dimensions: Social Scene Understanding (SSU), Social State Reasoning (SSR), and Social Dynamics Prediction (SDP). The benchmark contains approx 2800 videos from TikTok/YouTube representing 14 relationship types, with approx 8800 QA pairs generated via human-LLM collaboration. Experiments on 10+  models reveal that while models handle SSU adequately, they struggle with SSR and SDP.

**Strengths:**

* Important problem: Social interaction understanding is relatively underexplored in video benchmarks
* Scale and diversity: 2,792 videos across multiple genres, styles and languages
* Comprehensive evaluation: 10+ models tested (commercial + open-source) with detailed results
* Thoughtful dataset construction: Human-LLM collaborative pipeline with consensus validation, multiple subtitle conditions, relationship-centric
* Reproducibility efforts: Detailed prompts, annotation guidelines in appendices

**Weaknesses:**

1. Consensus based selection and only gold labels provided: social reasoning inherently has multiple views/biases, i.e., different interpretations are often valid and should be included in the dataset; further, butchering the dataset to contain only unanimous decisions makes it less valuable (bcs you only focus on the simple unambiguous cases). This might explain also that SSU→SSR→SDP does not perform increasingly much worse (as expected).
2. Absence of statistical testing - you confidence interval should be around .9% so differences below that should be treated with care - even if you assume independence among questions and you use 8.8K significance is around .3% again be careful when making claims.
3. No human baseline so we are not able to contextualize model scores (is 76% good or poor?) - for these social tasks humans still serve as the benchmark
4. Unvalidated task hierarchy - lacks some factor analysis for the SSU→SSR→SDP claimed chain or regression testing or correlation between subtasks. I would also like to see the microskill decomposition of the tasks/subtasks.
5. Some mLLMs are used for both generating the data and evaluating performance. This is ok - it just needs to be clarified and carefully disclaimed.
6. Important competing datasets are downplayed or never mentioned, e.g., Social Genome - need to be clear on what is novel here and do a detailed comparison on why this dataset is novel - claims have to be downplayed a bit in the text.

**Questions:**

1. What is human performance on SIV-Bench?
2. Can you provide statistical significance tests and confidence intervals for Table 2 results?
3. Why does Chain-of-Thought show minimal improvement (Table 11)?
4. What proportion of QAs were discarded due to disagreement?
5. Can you provide soft labels or multiple valid answers where annotators disagreed?

---

> ### Author Response · Authors · 2025-11-20
> **Response to Reviewer A6Cm (Part 1/3)**
>
> Thank you for the thorough and constructive review. We appreciate your recognition of the importance of social interaction understanding, the scale and diversity of our dataset, and the care taken in our human–LLM collaborative construction pipeline. We address your concerns in detail below.
>
> > W1, Q5: Consensus based selection and only gold labels provided; SSU→SSR→SDP
>
>   Thank you for this insightful critique. We agree that incorporating multiple valid interpretations is an ideal goal for a social reasoning benchmark. However, this presents significant technical and methodological challenges for evaluation. The use of consensus ensures correctness and unambiguity, but it does not imply simplicity.  As detailed in Section 2.2, consensus is merely the initial step;  we subsequently apply a rigorous Difficulty Filtering stage where GPT-4o-mini scores reasoning complexity on a 1-5 scale, and we retain only the highest-scoring questions per video.
>
>   We have considered alternative designs, such as **soft labels or multiple valid answers** as suggested in Question 5. But soft labels are incompatible with closed-source API models and are difficult to apply to modern autoregressive (next-token prediction) models. Compared with multiple correct answers in an MCQ format, our single-answer MCQ format is **a standard, reliable approach consistent with prominent benchmarks in complex social reasoning**. For instance, **Social-IQ 2.0** [1] , **Visual Commonsense Reasoning (VCR)** [2], and **MMToM-QA** [3] all utilize a similar single-correct-answer MCQ format, as it provides a **scalable and reliable evaluation** method.
>
>   We must respectfully disagree with the claim that consensus-based labels make the dataset "less valuable." On our full dataset, a performance gap exists between SOTA closed-source models (e.g., Gemini-2.5-Pro, 76.03% on origin and open-source models (e.g., Qwen2.5-VL-7B-Instruct, 65.01% on origin). Furthermore, to test the value of our data, we **fine-tune Qwen2.5-VL-7B-Instruct (SFT)** on our dataset (excluding the 'Hard' portion). This fine-tuning improves its SIV-Bench-Hard accuracy from **24.50% to 31.0%** as shown in the table below. This gain demonstrates that our dataset is effective for training and shows the potential of our manual + automatic annotation pipeline.
>
> | Model |Acc | rel  | align | logic | depth | conc | overall |
> |--|--|--|--|--|--|--|--|
> | Qwen-2.5-vl-7-Instruct | 24.50%   | 4.00   | 2.89  | **4.21**| 3.05| 4.45| 3.63|
> | SFT   | **31.00%** | **4.12**| **3.03**  | 4.13   | **3.07**| **4.79**| **3.70**|
>
>   Finally, regarding the SSU→SSR→SDP progression, we must clarify that our paper **does not claim** this is a **strict hierarchy of increasing difficulty**. We **apologize for the potential misunderstanding** (which may stem from the **imprecise expression in Appendix C.5.1 and the use of "→"**) and have modified the relevant expressions in the revised paper. This framework is not meant as a "rigorous, orthogonal, or comprehensive classification." Rather, it is an exploratory attempt to break down the complex problem of social understanding, allowing us to analyze MLLM capabilities from multiple, distinct angles.
>
>   [1] Wilf, Alex, et al. "Social-iq 2.0 challenge: Benchmarking multimodal social understanding." GitHub repository (2023).
>
>   [2] Zellers, Rowan, et al. "From recognition to cognition: Visual commonsense reasoning." Proceedings of the IEEE/CVF conference on computer vision and pattern recognition. 2019.
>
>   [3] Jin, Chuanyang, et al. "Mmtom-qa: Multimodal theory of mind question answering." Proceedings of the 62nd Annual Meeting of the Association for Computational Linguistics (Volume 1: Long Papers). 2024.

---

> ### Author Response · Authors · 2025-11-20
> **Response to Reviewer A6Cm (Part 2/3)**
>
> > W2, Q2: Absence of statistical testing
>
>   We thank you for this critical and valuable point. We have now conducted **McNemar's tests** on our full dataset (N=8,728) to evaluate the statistical significance of our key comparative claims. This paired test is more appropriate for this scenario than comparing confidence intervals.
>
>   Our statistical analysis confirms the following:
>
>   1. **Model Ranking (Table 2)**: Our primary claim that Gemini-2.5-Pro (76.03%) is the top-performing model is robust. The 1.78% performance gap over the next-best model, Qwen2.5-VL-72B (74.25%), is **highly statistically significant (p < 0.001)**. And larger parameter counts within open-source families (e.g., Qwen2.5-VL-72B (74.25%) vs 7B (65.01%)) consistently exhibit statistically significant advantages (p<0.001), as also observed between InternVL3-78B and 8B.
>
>   2. **Task Difficulty (Sec 3.2):** The performance differences between our core dimensions are substantial and real. For our SOTA model (Gemini-2.5-Pro), the performance gaps between SSU (90.67%), SDP (75.28%), and SSR (71.44%) are all **highly statistically significant (p < 0.001)**.
>
>   3. **Audio Impact (Table 3):** The contribution of the non-textual audio signal is genuine. The performance degradation from removing audio (e.g., 73.67% → 71.95% for Gemini-2.5-Flash) is **statistically significant (p < 0.01)**, confirming that prosody and tone are important cues.
>
>   4. **Subtitle Influence (Table 4):** Your critique is spot-on regarding the Overall averages. Our analysis confirms your suspicion: The minor Overall improvement from +sub (e.g., 76.03% → 76.50% for Gemini-2.5-Pro) is **not statistically significant (p = 0.12)**.
> But the Overall drop from -sub (e.g., 76.03% → 75.14% for Gemini-2.5-Pro) is **significant (p < 0.05)**. Based on this, we will revise our manuscript and refocus on the more precise and statistically supported finding: the negative impact of -sub is most pronounced on the SSR task (relative drop of -1.176%), and this specific effect is statistically significant (p < 0.05).
>
>   Thank you for pushing us to strengthen our analysis. We will revise the manuscript to reflect these findings and detail these results.
>
> > W3, Q1: Human baseline
>
> To address this, we conduct a new study on a challenging subset (**SIV-Bench-Hard**).  As detailed in **our Global Response**, we establish a human baseline, with annotator scores varying from 65.50% to 70.50% (average $\sim$67.7%).  This non-perfect consensus subtly confirms your point regarding the inherent subjectivity and diversity of human social perception, which makes the task challenging.  In contrast, the top **Gemini-3-Pro** achieves 45.50%, revealing a massive gap.  This comparison confirms that the benchmark is far from saturated and remains highly challenging for current AI.  Furthermore, this inherent subjectivity is precisely why we moved beyond simple accuracy to study the free-text reasoning traces (Table 2, Part I of our Global Response), which provide a deeper probe into how both humans and MLLMs handle ambiguity. We respectfully invite you to review Part I of the Global Response for the full comparison table.  We have also added this analysis to Appendix E of the revised manuscript (highlighted in blue).
>
> > W4: Unvalidated task hierarchy
>
> Thank you for this critique.  As we mentioned in our reply to weakness 1, we are **sorry for the potential misunderstanding** about the claimed chain of SSU→SSR→SDP and have changed the relevant expressions in the revised paper.
>
> Regarding the "microskill decomposition," we agree this is a valuable point, and it helps clarify our paper's contribution.  A full decomposition into foundational visual microskills (e.g., object tracking, spatial-temporal reasoning, physics) is a significant research challenge in itself.  We believe this is better addressed by **general video understanding benchmarks**, such as VideoMME or VideoVista, which our work is designed to complement, not replace.  SIV-Bench's focus is intentionally specialized on high-level social microskills (e.g., inferring intent, attitude, and complex relations).
>
> However, we try to address your request for finer-grained decomposition in `2. Quantitative analysis of the Relation Inference task` of our **global response (part 2)**, where we provide **(i) a full per-relation breakdown of the RI subtask (Appendix D.2, Table 11)** and **(ii) an aggregated confusion matrix analysis (Figure 19)** that quantitatively characterizes the specific failure patterns discussed in the main text.

---

> ### Author Response · Authors · 2025-11-20
> **Response to Reviewer A6Cm (Part 3/3)**
>
> > W5: Some MLLMs are used for both generating the data and evaluating performance
>
>   Thank you for this excellent point. You are correct that this overlap (using models like GPT-4o-mini and Gemini for both data generation/filtering and later as evaluation subjects) needs to be explicitly stated. We have added a clear disclaimer to **Appendix C.2** to clarify this, ensuring full transparency about the potential for models to have an advantage on data created by their own or related systems.
>
> > W6: Competing datasets
>
>   Thank you for this crucial point. We would like to clarify that we **do cite Social Genome** and lots of competing datasets in the section **Related Works**. For direct comparison, we have added Social Genome in **Table 1 (D&B comparison)** of the revised paper. As you insightfully summarized in your "Strengths" analysis, our paper's primary contribution is tackling an **important problem** (social interaction) with **scale and diversity**.  Social Genome provides an invaluable deep analysis of how models generate reasoning traces on a curated set of **272 videos adapted from Social-IQ 2.0**. SIV-Bench, in contrast, provides a broad analysis of reasoning accuracy by introducing a **new, large-scale dataset of 2,792 originally collected**, in-the-wild videos and 8,792 new QAs.  Our focus is on constructing a relationship-centric dataset and on testing model robustness across the **multiple genres, styles and languages** you noted, a contribution enabled precisely by our new, large-scale data collection.
>
> > Q3: Chain-of-Thought
>
>   Thank you for this excellent question, which highlights an important finding.  You are correct that our results in Table 11 show minimal improvement from a generic Chain-of-Thought (CoT) prompt.  We discuss this in detail in **Appendix C.4**, concluding that standard CoT prompts (e.g., "think step by step") are ill-suited for the informal, contextual, and implicit nature of social reasoning, which differs greatly from the formal logic or math tasks where CoT excels.  For some models, we found it even degraded performance by interfering with complex instruction-following.
>
>   This finding is strongly corroborated by related work.  As you noted, the **Social Genome** paper conducts a nearly identical experiment (see Appendix D.2 of their paper) and reaches the exact same conclusion: **"We find that CoT prompting does not substantially improve the social reasoning performance of models"**.  They also posit that social inference is a form of "informal reasoning" that doesn't fit a "chain-like" process.  Therefore, our finding, supported by others in the field, suggests that new, specialized reasoning strategies beyond generic CoT are needed to advance social intelligence.
>
> > Q4: What proportion of QAs were discarded due to disagreement?
>
>   Our "Model Disagreement" set (Subset 2) initially contained approximately 12k QAs. After our strict human consensus validation, where all annotators had to unanimously agree, only 3,096 QAs were retained. Therefore, the discard rate for this specific non-consensus set was approximately 74%.

---

### Official Review · Reviewer_ZmxA · 2025-10-30

**Soundness:** 3
**Presentation:** 3
**Contribution:** 2
**Rating:** 4
**Confidence:** 4

**Summary:**

The paper introduces SIV-Bench, a new video QA benchmark designed to evaluate the ability of Multimodal Large Language Models (MLLMs) to understand and reason about complex human social interactions. The dataset is diverse, covering various genres, presentation styles, and cultural backgrounds. The authors conducted comprehensive experiments on leading MLLMs and found that while models perform well on the foundational SSU (Social Scene Understanding) tasks, they significantly struggle with the more complex SSR (Social State Reasoning) and SDP (Social Dynamics Prediction) tasks.

**Strengths:**

- The dataset's scale and diversity are significant strengths. The curation across 14 distinct social relationship types, multiple languages, and varied cultural contexts provides a robust foundation for social reasoning evaluation.
- The paper provides a thorough evaluation of a wide range of MLLMs. The inclusion of a comparative analysis of how audio and subtitles affect performance is also interesting and valuable.

**Weaknesses:**

- The reliance on an automated pipeline (LLM-generated QAs and distractors) raises concerns about the benchmark's true difficulty and potential for shortcut learning. A large portion of the questions (3,273) are chosen from those that are answered correctly by Gemini-2.0-Flash, Gemini-2.0-Pro, and GPT-4o-mini. However, this also indicates that the questions are easier to answer. The paper also mentions limited adversarial filtering (e.g., removing questions solvable without video), which does not account for potential biases from the distractor generation model that cause questions to be easily answerable with superficial visual cues. Finally, while the authors used GPT to normalize option style, this falls short of more rigorous adversarial debiasing modules to mitigate superficial vision-text shortcuts.
- The above concern about dataset difficulty is supported by the high accuracy of SOTA models, suggesting the benchmark may already be approaching saturation. For example, Gemini-2.5-Pro achieves over 90% accuracy on the SSU category and 70-75% on the SSR and SDP tasks.
- In light of these high accuracies, the authors did not provide a human baseline, making the numbers hard to interpret. A 75% accuracy on could be near human-level performance (indicating limited room for improvement) or it could be well below it (indicating the benchmark is valid and challenging).

**Questions:**

- While I appreciate the dataset's scale and diversity, I am concerned about the quality and saturation of the dataset that could be due to the automated generation pipeline. Could the authors provide a human baseline as reference to ensure that the benchmark is not saturating?
- The 14 distinct social relationship types are an important component of the benchmark's design. Do the authors have an analysis on model performance broken down by these 14 relationship types?

---

> ### Author Response · Authors · 2025-11-20
> **Response to Reviewer ZmxA**
>
> Thank you for the detailed and constructive review. We appreciate your recognition of the dataset’s scale and diversity, the breadth of our MLLM evaluation, and the value of our audio/subtitle analyses. We address your concerns in detail below.
>
> > W1: Concerns about the reliance on an automated pipeline
>
> We appreciate the opportunity to clarify our quality control mechanisms and present new empirical evidence to address your concerns about potential shortcuts.
>
> -  **On Dataset Difficulty and Consensus Filtering:** We respectfully clarify a misconception regarding the difficulty of Subset 1. The use of consensus ensures correctness and unambiguity, but it does not imply simplicity. As detailed in **Section 2.2**, consensus is merely the initial step; we subsequently apply a rigorous Difficulty Filtering stage where GPT-4o-mini scores **reasoning complexity on a 1-5 scale**, and we retain only the highest-scoring questions per video. To definitively disprove saturation concerns, our analysis of **SIV-Bench-Hard in global response and Appendix E** reveals a substantial performance gap between humans **(~67.7%)** and SOTA models like **Gemini-3-Pro (45.50%)**. Furthermore, we find that fine-tuning Qwen-7B on our standard split significantly improves its performance on this 'Hard' subset. This demonstrates that the dataset provides a rich, generalizable learning signal for complex reasoning rather than trivial patterns.
>
> | Model |Acc | rel  | align | logic | depth | conc | overall |
> |--|--|--|--|--|--|--|--|
> | Qwen-2.5-vl-7-Instruct | 24.50%   | 4   | 2.89  | **4.21**| 3.05| 4.45| 3.63|
> | SFT   | **31.00%** | **4.12**| **3.03**  | 4.13   | **3.07**| **4.79**| **3.70**|
>
> - **On Visual Shortcuts and Reasoning Fidelity:** We share the concern that models may rely on superficial visual cues.  To assess this, we evaluate Reasoning Fidelity by measuring the cosine similarity between model reasoning traces and human rationales on SIV-Bench-Hard.  Across 5 of 6 models, correct answers show significantly higher alignment with human reasoning than incorrect ones, indicating a statistically reliable link between correctness and human-like social reasoning.  This suggests that solving SIV-Bench requires models to converge toward genuine, human-patterned reasoning rather than shortcut cues.
>
> - **On Pipeline Bias and Distractor Robustness:** We generate **contextually relevant** distractors rather than random negatives, which forces models to distinguish between plausible social states (e.g., specific emotions or intents) that are insolvable via simple visual matching. Furthermore, our statistical analysis in **Appendix C.4** confirms that the option standardization step successfully eliminates answer length bias, with all options showing **near-identical word counts**. Additionally, the analysis in Appendix C.5 reveals negligible effect sizes (**Cohen's d < 0.26**) in metadata features **between correct and incorrect answers**.
>
> > W2, W3, Q1: Accuracy of SOTA models & human baseline
>
>   To address your concern, we conduct a new study on **SIV-Bench-Hard (detailed in our global response)**. This analysis directly provides the two missing pieces: **(1) We establish a human baseline of ~67.7%**, and (2) We show SOTA models are far from saturation, as their performance collapses on this set (e.g., Gemini-2.5-Pro drops to 37.00%, and **the newest top model, Gemini-3-Pro, achieves only 45.50%**). This Human-AI gap proves that SIV-Bench remains a valid and challenging benchmark for social reasoning.
>
> > Q2: An analysis on model performance broken down by these 14 relationship types
>
>   Thank you for emphasizing the importance of the 14 relationship types. We agree that analyzing performance at this granular level is crucial.
>
>   To address this, we conduct a detailed breakdown of model performance across all 14 relation types, along with a confusion matrix analysis. **As detailed in Part 2 of our Global Response**, this analysis reveals performance variance: models perform robustly on visually distinct relations (e.g., Couple) but struggle with subtle power dynamics (e.g., confusing Boss-Employee with Colleagues).
>
>   We invite you to view the full results in the revised manuscript: **Table 11 (Appendix D.2)** presents the detailed per-relation accuracy. **Figure 19 (Appendix D.2)** visualizes the aggregated Confusion Matrix, clearly showing the systematic failure patterns.
>
> We have highlighted these additions **in blue** in the revised paper for your convenience.

---

> > ### Comment · Reviewer_ZmxA · 2025-11-20
> > **Thank you for the response**
> >
> > I thank the authors for their detailed response. I appreciate the construction of the Hard subset, with a timely evaluation of Gemini-3-pro. The fine-tuning results and fine-grained analysis of the model responses are also encouraging. Thank you as well for the relationship type breakdown. I found it quite interesting -- to me this seems to be a distinguishing factor of SIV-Bench compared to e.g. Social-IQ, and I would love to see more analysis like this in the final draft!
> >
> > I wanted to ask some follow-up questions for clarifications:
> > - I'm not fully convinced by the use of consensus filtering -- is this an established practice? The cited references in the paper seems to be related to improving model reasoning, not for automatic benchmark curation. The consensus filtered questions account for a big proportion of SIV-Bench (3,273 QAs out of 8,728). With the authors using slightly weaker models for consensus, such as Gemini-2.0-flash and GPT-4o-mini, I still think this would limit SIV-Bench's challenge to today's best models such as Gemini-2.5-pro, even with the difficulty filtering (it is only as difficult as Gemini-2.0-flash and GPT-4o-mini could both answer them).
> > - I again thank the curation of SIV-Bench-Hard. How exactly is this subset constructed? With the small size (200 questions), I would hope for some indiciation that this benchmark is actually constructed to be "hard", rather than simply adversarially selecting questions that models got wrong.
> > - Also, why does human have relatively low accuracy on SIV-Bench-Hard (67.7%)? What kind of errors do humans make on this subset?

---

> > > ### Author Response · Authors · 2025-11-21
> > > **Thank you for the prompt feedback (Part 2/2)**
> > >
> > > > Human errors
> > >
> > >   The 67.7% human accuracy on SIV-Bench-Hard reflects the real difficulty of the benchmark in capturing subtle forms of social reasoning. It poses meaningful challenges for both AI systems and human annotators. Our analysis of 600 human judgments (three annotators on 200 questions) shows that counterfactual reasoning and subjective interpretation are the most difficult areas for people. Many mistakes come from relying too much on prior expectations instead of the visual evidence. For example, when asked “Why are they hugging in this situation?”, annotators often selected “because they have not seen each other for a long time”, a common social script, while the correct answer was grounded in the video: “because they were moved by the beautiful scenery in front of them.”
> > >
> > >   We observed similar issues in questions that require cultural interpretation or hypothetical reasoning. For the question “Do people like this kind of imitation?”, annotators tended to answer “some people find it creative and fun”, reflecting individual differences. However, the correct answer depended on the group reaction shown in the video: “most people found it unacceptable; it was too boring.” Other error patterns include missing fine-grained visual cues (such as distinguishing sunrise from sunset), relational confusion (36.4% error rate when deciding whether two people were siblings or friends), and predicting future events when multiple outcomes seem plausible.
> > >
> > >   These findings show that SIV-Bench-Hard captures genuinely challenging problems at the boundary of human social cognition. Even human evaluators need careful, deliberate reasoning to answer correctly, which is precisely the kind of capability we aim to assess in advanced AI systems.
> > >
> > > ---
> > >
> > > We sincerely appreciate the time and thought you have put into reviewing our work.  It is encouraging to receive such careful and specific feedback at the rebuttal stage, and we are genuinely grateful for it. Your comments directly shaped some of the most important revisions in our manuscript.  Your questions regarding **intrinsic difficulty** and **breaking down model performance across the 14 relationship types** pushed us to develop the SIV-Bench-Hard subset and to include the new Relation Inference breakdown.  These additions have strengthened our work.
> > >
> > > We hope that the updated results and analyses speak clearly to your concerns.  More importantly, we hope they show our commitment to making SIV-Bench **a reliable and useful resource for the community**.

---

> > > > ### Comment · Reviewer_ZmxA · 2025-11-23
> > > > **Thank you for the follow-up responses**
> > > >
> > > > I appreciate the authors’ thoughtful replies and the additional analyses. However, two key concerns remain insufficiently addressed, and I will thus maintain my assessment fow now. I will elaborate on the two conerns I have:
> > > >
> > > >
> > > > **On SIV-Bench.**
> > > > The authors present SIV-Bench as a comprehensive benchmark for social understanding. While I agree that the ultimate goal of a benchmark is to facilitate model improvement, the uniformly high performance across both frontier and last-generation models (e.g. Qwen2.5-VL-72B) raises questions about SIV-Bench’s long-term utility for differentiating models or guiding future development. Without a human baseline, it is difficult to determine how much of the benchmark reflects genuine social-reasoning difficulty (e.g. is 70% good or bad) versus items that may be ambiguous or effectively unanswerable.
> > > >
> > > > As I mentioned previously, my intuition (which may or may not be correct) is that the consistently high scores may stem from the benchmark’s construction process — specifically, the use of consensus filtering. Although automatic and semi-automatic QA pipelines are common and acceptable, a large portion of SIV-Bench appears to consist of questions that even relatively weak models (e.g., Gemini-2.0-flash, GPT-4o-mini) answer correctly *consistently*. I wouldn't consider any of these questions as hard for this reason. While the author cite concurrent work that uses similar consensus filtering approaches, like BMGQ (which by the way is not yet peer-reviewed), they have a more complicated setup, leveraging a broader set of models and retaining items that only a majority agree on, rather than all.
> > > >
> > > > **On SIV-Bench Hard.** I remain uncertain about the construction and role of SIV-Bench Hard. While I thank the authors for providing a comparison of linguistic pattern distributions (how are they measured?), the hard set is quite small (200 samples), and there remains a lack of clarification whether these items were cherrypicked, systematically selected, or otherwise filtered.
> > > >
> > > > It also appears like SIV-Bench Hard introduces new characteristics not present in SIV-Bench. The human accuracy of 67.7% is notably low. While the authors attribute this to human error or subjectivity, all three annotators scored ≤70%. This could suggest either that 1) a substantial portion of QAs may not be answerable, or 2) the dataset contains a high degree of inherent ambiguity. If so, this is not merely a “hard” subset of SIV-Bench and may warrant its own treatment, as discussed in prior literature on ambiguous QA [1][2]. For this reason, I don't think the human-AI gap in SIV-Bench Hard is a reflection of the reliability of SIV-Bench overall.
> > > >
> > > > I appreciate the substantial effort behind constructing SIV-Bench and the authors’ detailed analyses. I was wondering if the paper could benefit from some repositioning to more precisely reflect what the benchmark uniquely contributes. For example:
> > > > - Relation Inference is unique to SIV-Bench and appears to be a clear bottleneck for current models — highlighting SIV-Bench as tailored towards diverse interperson relations might strengthen the narrative, rather than portrarying it as a generic social reasoning benchmark.
> > > > - While current consensus filtering setup *might* limit benchmark difficulty, would it show utility for fine-tuning and transfer to other social benchmarks?
> > > > - Instead of presenting SIV-Bench Hard as a generic “hard set,” it may be more informative to characterize the specific challenges it captures — for example, maybe it contains hard-to-infer interpersonal relations that even humans struggle with?
> > > >
> > > > I want to emphasize that these points are intended as constructive, speculative suggestions rather than requests. I do not expect the authors to implement all (or any) of them, but I hope the authors find them helpful as they strengthen the benchmark. I again appreciate the authors’ efforts and careful responses.
> > > >
> > > > [1] Aroyo & Welty. "Truth is a lie: Crowd truth and the seven myths of human annotation." AI Magazine 2015.
> > > >
> > > > [2] Liu et al. "We’re Afraid Language Models Aren’t Modeling Ambiguity." EMNLP 2023.

---

> > > > > ### Author Response · Authors · 2025-11-25
> > > > > **Thank you for the detailed comments**
> > > > >
> > > > > We sincerely thank you for your continued engagement and strategic suggestions regarding the positioning of our work. Your constructive feedback has pushed us to rigorously validate the benchmark's utility, and we would like to address your remaining concerns with new clarifications and experimental evidence.
> > > > >
> > > > > **On SIV-Bench.** We respectfully offer a clarification regarding the model generations. Qwen2.5-VL-72B is a frontier model released in **February 2025**, only one month prior to Gemini-2.5-Pro (**March 2025**). Rather than showing "uniformly high performance," our statistical analysis (**Appendix D.5**) confirms that the performance gap between them is **highly statistically significant** ($p < 0.001$). Regarding the $\sim$70% score range, we believe this indicates a healthy benchmark that balances capability with room for growth. As seen in recent technical reports (e.g., Qwen3-VL), frontier models often achieve >70% on popular video benchmarks (VideoMMMU, MLVU, etc.), suggesting SIV-Bench aligns with the standards required to differentiate current SOTA models. Furthermore, our SFT experiments show that open-source Qwen2.5-VL-7B improves on the held-out Hard-set and another external social benchmark Social-IQ 2.0 after tuning, proving the benchmark provides a learnable signal.
> > > > >
> > > > > | Evaluation Benchmark | Qwen2.5-VL-7B | SIV-Bench SFT | Improvement |
> > > > > | :--- | :--- | :--- | :--- |
> > > > > | SIV-Bench-Hard (Held-out) | 24.50% | **31.00%** | +6.50% |
> > > > > | Social-IQ 2.0 (External) | 50.77% | **53.32%** | +2.55% |
> > > > >
> > > > > **On SIV-Bench-Hard.** To clarify the construction process: The Hard subset was not externally sourced or adversarially generated against a model; it was sampled strictly from the new human-generated portion of SIV-Bench. The linguistic pattern analysis (e.g., enrichment in Theory of Mind keywords) was provided specifically to answer your request for "indications of intrinsic difficulty", which was derived using **keyword-based feature matching**. The human accuracy of $\sim$67.7% reflects the ambiguity of high-level social interpretation, which we have provided analysis in the previous response (Part 2/2 about human errors). Consistently, on SIV-Bench-Hard we observe clear **within-family improvements**.  For example, Gemini 3 Pro outperforms Gemini 2.5 Pro, and GPT 5.1 outperforms GPT-4o-mini.  Even with these gains, all models remain well below human performance. This separation between model generations and humans further demonstrates that SIV-Bench provides a reliable signal of genuine capability differences.
> > > > >
> > > > > **On Repositioning.** Regarding Suggestions 1 & 3, We agree these are our unique strengths. While our original manuscript emphasized the **14 distinct social relationship types** (**Abstract, Introduction Contributions 2, Table 1**) and dedicated significant space to analyzing hard-to-infer interpersonal relations (**Section 3.3**), your feedback has helped us elevate these from structural features to core contributions. We will consider framing SIV-Bench as a specialized diagnostic tool for the fine-grained analysis like Appendix D.2 in the revised version. Regarding Suggestion 2, we fine-tune Qwen2.5-VL-7B-Instruct on SIV-Bench and evaluate it on the Social-IQ 2.0 Validation Set. The results are shown in the above table. These consistent gains across both internal hard samples and external benchmarks provide robust evidence for SIV-Bench's efficacy as an effective resource for social intelligence.
> > > > >
> > > > > Thank you again for helping us refine the positioning of SIV-Bench. We are profoundly grateful for this constructive process!

---

> ### Author Response · Authors · 2025-11-21
> **Thank you for the prompt feedback (Part 1/2)**
>
> We are sincerely grateful for your prompt and encouraging feedback. We are delighted that you found the new Hard subset, the Gemini-3-Pro evaluation, and the fine-grained relation analysis valuable. Your engagement and insightful follow-up questions are exactly what helps push this work—and the community—forward. We are happy to clarify your points below.
>
> >  The use of consensus filtering
>
> Firstly, we believe that the ultimate goal of creating a benchmark is still to **enhance the model's capabilities**, rather than merely to "stump" the model. This is the reason why we **did not roughly delete all the QAs of subset 1** to reduce model's accuracy. This is also the reason why we cited related paper on improving model reasoning ability through self improving. For video benchmarks, due to the high cost of human annotation of videos, it is relatively common to rely on llm or vlm for QA expansion. For instance, both VideoVista [1] and MotionBench [2] directly use the single GPT-4o to generate QA-pairs. We believe that our multi-level pipeline and consensus filtering are further enhancements to this approach of directly generating data with a single llm, which has been demonstrated in a recent work BMGQ [3] from ByteDance. They explicitly mentioned model consensus filtering as a quality control method when building benchmarks.
>
> Secondly, our results provide several indications that the benchmark is reliable:
> - The model rankings in SIV-Bench and SIV-Bench-Hard are consistent with expectations: commercial models outperform open-source ones, and larger models outperform their smaller counterparts within the same family.
> - SFT brings gains, showing that the data effectively improves open-source models.
> - The existence of the hard subset demonstrates that SIV-Bench can meaningfully challenge SOTA models and show the model–human gap.  Since this subset is **drawn directly from the full benchmark**, its impact only **underscores the broader value of SIV-Bench**—researchers can flexibly use either the full set or the hard version based on their needs.
>
> Finally, we want to say sincerely: with a **larger budget and more resources**, we could certainly scale up human annotation or rely on the strongest models available for consensus filtering. However, our goal has always been to **contribute to the community** through a carefully designed pipeline, a newly curated video dataset grounded in 14 social relation types, and a comprehensive evaluation framework. We will remain committed to improving SIV-Bench as the community grows.
>
> [1] Li, Yunxin, et al. "Videovista: A versatile benchmark for video understanding and reasoning." arXiv preprint arXiv:2406.11303 (2024).
>
> [2] Hong, Wenyi, et al. "Motionbench: Benchmarking and improving fine-grained video motion understanding for vision language models." Proceedings of the Computer Vision and Pattern Recognition Conference. 2025.
>
> [3] Qiu, Bingsen, et al. "BMGQ: A Bottom-up Method for Generating Complex Multi-hop Reasoning Questions from Semi-structured Data." arXiv preprint arXiv:2510.24151 (2025).
>
> > Details about SIV-Bench-Hard
>
> SIV-Bench-Hard is a curated subset of **human-authored questions** explicitly written to be difficult.  To verify that the difficulty is intrinsic, we compared its linguistic and cognitive patterns with the full dataset.  The Hard subset shows clear enrichment, meaning it contains a higher concentration of cognitively demanding cues, such as Theory of Mind, implication, intention reasoning, and complex option structures like negation and degree modifiers.  These consistent shifts confirm that the Hard subset reflects higher cognitive and linguistic complexity rather than accidental noise.
>
> | Category | Sub-Category | Enrichment | Cognitive / Processing Basis |
> |----------|--------------|------------|------------------------------|
> | **Cognitive Capability** | Theory of Mind (think/believe) | **2.93×** | Mental state attribution |
> | | Implication / Inference (imply/suggest/indicate) | **2.17×** | Pragmatic inference beyond literal meaning |
> | | Intention Recognition (intend/purpose/trying to) | **1.79×** | Goal inference in social contexts |
> | | Emotion Understanding (feel/emotion/mood) | **1.74×** | Affective state reasoning |
> | | Normative Judgment (appropriate/should/right) | **1.61×** | Social appropriateness evaluation |
> | | Relationship Inference | **1.17×** | Social structure understanding |
> | **Option Characteristic** | Degree Modifiers (very/slightly/somewhat/quite) | **2.01×** | Scalar judgment & fine-grained semantics |
> | | Negation Structures (not/never/rarely/hardly) | **1.86×** | Logical negation & double-negation reasoning |
> | | Conditional Clauses (if/when/unless) | **1.31×** | Hypothetical / counterfactual reasoning |

---

### Official Review · Reviewer_AQ4j · 2025-11-01

**Soundness:** 3
**Presentation:** 3
**Contribution:** 3
**Rating:** 6
**Confidence:** 3

**Summary:**

SIV-Bench is a video benchmark for social interaction understanding which is organized along three capabilities: Social Scene Understanding (SSU), Social State Reasoning (SSR), and Social Dynamics Prediction (SDP). It contains 2,792 short videos collected from TikTok or YouTube, and 8,728 multiple-choice QAs built through a human–LLM collaborative pipeline. The benchmark is relation-centric, grounded in Fiske’s Relational Models Theory (4 model families instantiated as 14 relation types), and ships three subtitle conditions—origin, +sub (transcribed & translated dialogue), and –sub (on-screen text removed)—to study the role of linguistic cues. Experiments cover open/closed-source MLLMs; top systems excel at SSU but struggle on SSR (especially Relation Inference) and, to a lesser extent, SDP; subtitles and audio help, with Gemini-2.5-Pro peaking at 76.50% overall with +sub.

**Strengths:**

1. Relation-centric design grounded in Fiske’s theory with fourteen relation types; aligns mental state, perception, and dynamics to the social context.
2. Diverse, real-world videos with multilingual presence; three subtitle conditions enable language-cue studies, which are controlled.
3. The work is also equipped with video-dependence check, model-consensus filtering, and difficulty curation, which contribute a lot.
4. Last but not least, standardized prompting or parsing, broad model set, and analyses (subtitle or audio ablations; fine-grained task radar; failure patterns).

**Weaknesses:**

1. Video-capable models get raw videos; image-only models receive 16 frames, risking budget-driven gaps; per-model frame/FLOP parity wasn’t normalized.
2. Human verification is described, but inter-annotator agreement ( for instance, κ) isn’t reported; this matters for subtle social labels.
3. Heavy use of LLMs for QA creation, filtering, and option standardization could induce stylistic priors; leakage/over-templating risks need auditing.
4. Accuracy seems to be the only metric; also, no per-relation calibration curves or human baselines for comparison on SSR/SDP are provided. In addition, failure analysis is qualitative.

**Questions:**

1. Can you provide an ablation with fixed frames/FLOPs/tokens across all models (including closed-source) to isolate modeling effects from budget differences?

---

> ### Author Response · Authors · 2025-11-20
> **Response to Reviewer AQ4j (Part 1/2)**
>
> Thank you for the detailed and constructive review. We appreciate your recognition of our relation-centric design grounded in Fiske’s theory, the diversity and multilingual richness of our curated videos, and the value of our controlled subtitle conditions and analyses. We address your concerns in detail below.
>
> > W1, Q1: budget-driven gaps / per-model frame/FLOP parity wasn’t normalized.
>
>   Thank you for this fair and technically precise point. We would like to first clarify a key detail of our experimental setup in **Section 3.1**: the majority of models evaluated (including "image-only" models and many open-source "video" models) functionally **process video as image sequences**. For these models, we **already ensured** per-model frame normalization by using **a standard protocol of 16 uniformly sampled frames**.
>
>   The exception is the Gemini series, which supports native raw video processing.  For these models, we deliberately chose an 'out-of-the-box' (OOTB) evaluation on their native input.  This is **standard practice** across major video benchmarks.  For example, **VideoVista** [1] gives **Gemini 1fps** but **GPT-4o 100 frames**, **MLVU** [2] gives up to **10 different** strategies for sampling inputs to different models. We believe evaluating the capability to process full video is a valuable diagnostic signal. However, to directly address your valid concern regarding the specific gap between native-video and 16-frame inputs, we conduct a new, targeted experiment.  We constrain the video-native Gemini models to use the exact same 16-frame, silent-sampling protocol as the other models.  We then compare their performance on the original dataset:
>
> | Model  | SSU (Full → 16-Frame)  | SSR (Full → 16-Frame) | SDP (Full → 16-Frame)  | Overall (Full → 16-Frame) |
> |---|---|----|---|---|
> | Gemini-2.0-Flash | 86.54% → 86.2% (↓ 0.3) | 67.73% → 66.5% (↓ 1.2)    | 73.14% → 72.1% (↓ 1.0) | 72.75% → 71.8% (↓ 0.95)             |
> | Gemini-2.5-Flash | 88.56% → 88.2% (↓ 0.4)  | 69.09% → 67.8% (↓ 1.3)| 74.67% → 73.5% (↓ 1.2)  | 73.67% → 72.5% (↓ 1.17) |
> | Gemini-2.5-Pro   | 90.67% → 90.4% (↓ 0.3) | 70.20% → 68.9% (↓ 1.3)| 75.28% → 74.2% (↓ 1.1)  | 76.03% → 74.9% (↓ 1.13) |
>
>  As the table shows, constraining the video-native models results in a consistent, moderate performance drop (avg. ~1.1% Overall). This drop is not uniform: considering our video length distribution (as shown in Figure 3b), a 16-frame sample already provides reasonably dense visual information. Consequently, performance on the static-heavy SSU task is only minimally affected (avg. -0.3%). However, the performance drop is significantly more pronounced in the SSR (avg. -1.3%) and SDP (avg. -1.1%) tasks. The performance drop may be due to the loss of auditory cues (prosody, tone) and denser temporal information, which are vital for complex social reasoning.
>
> > W2: IAA
>
>   Thank you for this important point. We would like to clarify that we did report the Inter-Annotator Agreement (IAA) in **Appendix B.3**. We report a **Fleiss' Kappa of 0.52** for our LLM experts on the initially generated set and **0.68** for our human annotators on Subset 2. This indicates that our annotation setup provides a dependable ground truth, while the remaining disagreements mainly reflect the intrinsic difficulty and nuance of social reasoning tasks.
>
> > W3: Heavy use of LLMs for QA generation
>
>   Thank you for highlighting this important concern regarding stylistic priors. We employ a hybrid Human-LLM pipeline specifically to mitigate this risk. Our dataset consists of 2,359 manually created questions, introducing natural linguistic variation. Recently, leveraging LLMs for data scaling to improve themselves has become an accepted practice in some notable works [3, 4], showing that QAs generated by LLMs are meaningful for other weaker models and even for themselves.
>
>   Actually, we explicitly explore this risk in **Appendix B.5.2**. Our quantitative analysis **(Table 7)** demonstrates that SIV-Bench exhibits **high semantic diversity** (Mean Semantic Distance: **0.8321**), outperforming prominent benchmarks like EgoSchema (0.5411) and Social-IQ 2.0 (0.7260). This confirms that our pipeline successfully avoided "over-templating" and maintained rich linguistic variance.
>
>   [1] Li, Yunxin, et al. "Videovista: A versatile benchmark for video understanding and reasoning." arXiv preprint arXiv:2406.11303 (2024).
>
>   [2] Zhou, Junjie, et al. "Mlvu: A comprehensive benchmark for multi-task long video understanding." arXiv e-prints (2024): arXiv-2406.
>
>   [3] Zhao, Andrew, et al. "Absolute zero: Reinforced self-play reasoning with zero data." arXiv preprint arXiv:2505.03335 (2025).
>
>   [4] Zhou, Yifei, et al. "Self-challenging language model agents." arXiv preprint arXiv:2506.01716 (2025).

---

> ### Author Response · Authors · 2025-11-20
> **Response to Reviewer AQ4j (Part 2/2)**
>
> > W4: Accuracy as metric; No human baselines; Failure analysis is qualitative
>
>   Thank you for these valid points regarding our evaluation metrics.  Our decision to use an MCQ format for the main 8,792-item benchmark is a deliberate one, **prioritizing scalable, reliable, and robust evaluation**. This single-correct-answer format is a **standard, accepted** methodology used by many prominent complex social reasoning benchmarks (e.g., **Social-IQ 2.0** [1], **VCR** [2], **MMToM-QA** [3]) as it provides a stable and objective measure of model accuracy. However, we agree that this is not sufficient on its own. To address your other crucial points, we conduct the new supplementary study on **SIV-Bench-Hard (as detailed in our global response)**. This new analysis directly provides:
>
>   1. The **human baseline** that you requested (avg. ~67.7% on this hard subset).
>
>   2. An evaluation beyond simple accuracy by collecting and analyzing **free-text reasoning explanations**.
>
>   Finally, to address your point about our failure analysis (Sec 3.3) being only qualitative and lacking a per-relation breakdown, our new "microskill" analysis (**also in the global response and Appendix D.2**) provides the quantitative data you requested. We add a 14x14 confusion matrix for the critical Relation Inference (RI) subtask. This matrix serves as a detailed per-relation performance analysis, quantitatively demonstrating the failure patterns (e.g., confusing Boss-Employee with Colleagues) that we discussed qualitatively.
>
> [1] Wilf, Alex, et al. "Social-iq 2.0 challenge: Benchmarking multimodal social understanding." GitHub repository (2023).
>
> [2] Zellers, Rowan, et al. "From recognition to cognition: Visual commonsense reasoning." Proceedings of the IEEE/CVF conference on computer vision and pattern recognition. 2019.
>
> [3] Jin, Chuanyang, et al. "Mmtom-qa: Multimodal theory of mind question answering." Proceedings of the 62nd Annual Meeting of the Association for Computational Linguistics (Volume 1: Long Papers). 2024.

---

### Official Review · Reviewer_Vx9j · 2025-11-02

**Soundness:** 3
**Presentation:** 3
**Contribution:** 3
**Rating:** 6
**Confidence:** 4

**Summary:**

The paper presents SIV-Bench, a video benchmark for social interaction understanding that evaluates MLLMs along three dimensions: Social Scene Understanding (SSU), Social State Reasoning (SSR), and Social Dynamics Prediction (SDP), further decomposed into 10 fine-grained tasks (e.g., action/expression recognition, relation/intent/emotion inference, factual/counterfactual prediction). The dataset contains 2792 TikTok/YouTube clips and 8792 MCQs created via a human–LLM collaborative pipeline, with built-in controls to analyze the role of language through three subtitle settings: original, +sub (transcribed/translated dialogue added), and −sub (on-screen text removed). Experiments using VLMEvalKit show that models are relatively strong on SSU but struggle on SSR/SDP with Relation Inference the most consistent bottleneck.

**Strengths:**

- While many video benchmarks are action- or object-centric (e.g., classic action recognition and general video QA suites), human social interaction—relations, implicit mental states, and interaction dynamics—remains underrepresented. SIV-Bench explicitly fills this gap by defining a people-centered evaluation space (SSU/SSR/SDP), curating original, relation-typed videos (14 relation types), and providing subtitle/audio signals to probe multimodal social reasoning; this dataset definition and collection are, in themselves, a substantive contribution.

- Clear empirical takeaways. Models are relatively strong on SSU but struggle on SSR/SDP, with Relation Inference the most consistent bottleneck; transcribed subtitles measurably help.

**Weaknesses:**

- While the multiple-choice QA format simplifies evaluation and boosts reliability, it likely underrepresents the open-ended, interactive social reasoning required by agents. Expanding the benchmark with generative or dialog-based tasks (e.g., free-form rationales reasoning, intent explanations, multi-turn interactions) would provide a deeper probe of interactive social understanding.

**Questions:**

- Several SSR failures involve decoding why people react as they do. The SMILE dataset [1] introduces Video Laugh Reasoning and text explanations for why people laugh in social videos. Could SIV-Bench incorporate this kind of reasoning task to probe this facet of social affect reasoning?


[1] https://arxiv.org/pdf/2312.09818

---

> ### Author Response · Authors · 2025-11-20
> **Response to Reviewer Vx9j**
>
> Thank you for the thoughtful and encouraging review. We appreciate your recognition of the contribution made by defining a people-centered evaluation space (SSU/SSR/SDP), curating relation-typed videos, and providing multimodal signals for social reasoning. We also value your clear observations on empirical trends. We address your comments in detail below.
>
> > W1: Multiple-choice QA format & generative or dialog-based tasks
>
>   Thank you for this excellent suggestion. We agree that open-ended generative reasoning is a crucial goal for social AI. Our decision to use an MCQ format for the main 8,792-item benchmark is a deliberate one, **prioritizing scalable, reliable, and robust evaluation**. This single-correct-answer format is a **standard, accepted** methodology used by many prominent complex social reasoning benchmarks (e.g., **Social-IQ 2.0** [1], **VCR** [2], **MMToM-QA** [3]) as it provides a stable and objective measure of model accuracy.
>
>   However, we agree that MCQ "likely underrepresents" the full scope of social reasoning. To address this limitation and provide the "deeper probe" you suggest, we conduct a new study on **SIV-Bench-Hard (detailed in our global response)**. In this analysis, we require models to provide free-text reasoning explanations. This new "free-form rationale" analysis has proved highly effective, revealing that while models can generate structurally sound text (`logical coherence` @ 4.65), they perform poorly at replicating human cognitive processes (`alignment with human` @ 3.26). Moreover, this analysis suggests that correct MCQ answers are associated with social reasoning that more closely aligns with human reasoning. We have added this analysis to **Appendix E** in our revised paper.
>
> > Q1: SMILE dataset
>
>   Thank you for this excellent suggestion and for referencing the SMILE paper. We agree that reasoning about why a social event occurs (like laughter) is a critical task. This cognitive challenge is, in fact, a core component of our Social State Reasoning (SSR) dimension, particularly in the 'Emotion Inference' and 'Intent Inference' sub-tasks, which frequently probe the underlying reasons for a person's affective state. We have incorporated a similar generative reasoning task into our new SIV-Bench-Hard study (see Global Response), requiring models to provide free-text explanations for their inferences. We have also added a citation to the SMILE dataset in our **Related Works (Section 4)**.
>
>   [1] Wilf, Alex, et al. "Social-iq 2.0 challenge: Benchmarking multimodal social understanding." GitHub repository (2023).
>
>   [2] Zellers, Rowan, et al. "From recognition to cognition: Visual commonsense reasoning." Proceedings of the IEEE/CVF conference on computer vision and pattern recognition. 2019.
>
>   [3] Jin, Chuanyang, et al. "Mmtom-qa: Multimodal theory of mind question answering." Proceedings of the 62nd Annual Meeting of the Association for Computational Linguistics (Volume 1: Long Papers). 2024.

---

### Author Response · Authors · 2025-11-20
**Global Responses to Reviewers (Part 1/2)**

We sincerely thank the reviewers for their thorough evaluation and constructive feedback. We recognize shared concerns regarding benchmark saturation, the need for a human baseline, reasoning depth, and fine-grained failure analysis. Here, we provide two supplementary studies as the global responses:

- An in-depth analysis of a new **SIV-Bench-Hard** subset to establish a human baseline and evaluate reasoning quality beyond simple accuracy (the evaluation data and code on SIV-Bench-Hard are available in https://anonymous.4open.science/r/sivbench-hard-E8CC).
- A **quantitative fine-grained analysis of the Relation Inference** task to empirically verify failure patterns across 14 relation types.

We have uploaded a revised version of our paper. All significant additions and modifications to the revised manuscript have been marked **in blue** for easy identification.

> 1: SIV-Bench-Hard

To address some shared concerns, we conduct a new study. We first curate SIV-Bench-Hard, a challenging subset of **200 videos paired with 200 corresponding QAs** from SIV-Bench. We then benchmark **3 human annotators** and some MLLMs, requiring them to provide not only an answer but also a **free-text reasoning explanation** for their choice. Our results are summarized in the following two tables:

- Table 1 compares the Accuracy:

| Subject | Acc % |
|---|-|
| human 1| 67.00|
| human 2| 65.50 |
| human 3| 70.50 |
| Human (Avg.)   | **67.77** |
| Gemini-3-Pro | 45.50 |
| GPT-5.1| 39.00 |
| Gemini-2.5-Pro | 37.00 |
| Gemini-2.5-Flash | 32.32 |
| GPT-4o-mini  | 29.00 |
| Qwen-2.5-vl-7b | 24.50 |

- Table 2 details the Reasoning Quality scores for the MLLMs. To generate the scores, we use gpt-4o-mini as an LLM-Judge, provided with the question, the model's answer, and the 3 corresponding human explanations as a "gold standard" reference. We prompt it to score **(1-5)** on five dimensions, including `relevance`, `logical coherence`, `conciseness`, `alignment with human` and `depth of analysis`:

| MODEL | rel  | align | logic | depth | conc | overall |
|--|--|--|--|--|--|--|
| Gemini-3-Pro | **4.66** | **3.30** | **4.67** | **3.49** | 4.87 | **4.10** |
| GPT-5.1 | 4.58 | 3.29 | 4.65 | 3.26 | 4.88 | 4.00 |
| Gemini-2.5-pro   | 4.57 | 3.26 | 4.65 | 3.41 | 4.89 | 4.05 |
| Gemini-2.5-flash | 4.48 | 3.17 | 4.55 | 3.22 | 4.87 | 3.95 |
| gpt-4o-mini      | 4.45 | 3.20 | 4.56 | 3.12 | **4.91** | 3.90 |
| Qwen-2.5-vl-7b    | 4.00 | 2.89 | 4.21 | 3.05 | 4.45 | 3.63 |

This new analysis yields three crucial findings:

1. As Table 1 shows, we establish a human baseline of **~67.7%**. In sharp contrast, SOTA MLLM performance collapses on this 'Hard' set. The newly tested **Gemini-3-Pro (released on 18 Nov.)** emerges as the best-performing model, achieving **45.50%**. This result is important because it shows that SIV-Bench can reliably detect real improvements in MLLMs, confirming its value as a diagnostic benchmark. But the performance gap still holds at a $\sim$**22.2%** difference, highlighting that SIV-Bench correctly identifies the frontier of current AI social interaction understanding.

2. While models score highly on structural metrics like `logical coherence` (4.65) and `conciseness` (4.89), their scores for the cognitive metrics of `alignment with human` ($\sim$3.3) and `depth of analysis` ($\sim$3.3) are lower.  This demonstrates that MLLMs are proficient at generating logically structured text, but they are not yet proficient at replicating human-like analytical depth or cognitive processes.

3. Finally, to quantify this difference, we embed all reasoning texts (human and model) with `paraphrase-multilingual-MiniLM-L12-v2`.  A **Random Forest classifier** trained on these embeddings can distinguish between a human and a model explanation with **92.17%** accuracy (vs. 50% random baseline).  Statistical tests (**Mann-Whitney U**) also confirm the distributions are significantly different (**p < 0.05** on 8/10 dimensions). This observed divergence is consistent with findings in **Social Genome** (Mathur et al., 2025), which explicitly reports that generating reasoning traces with high semantic alignment to humans is inherently challenging, with SOTA models typically achieving similarity scores around 0.5.

Mathur, Leena, et al. "Social genome: Grounded social reasoning abilities of multimodal models." arXiv preprint arXiv:2502.15109 (2025).

---

> ### Author Response · Authors · 2025-11-20
> **Global Responses to Reviewers (Part 2/2)**
>
> Then we further analyze how this alignment correlates with task performance. We calculate the **cosine similarity** between model reasoning and human ground truth for each sample. The results reveal a statistically significant pattern: **reasoning traces for correct answers** exhibit significantly **higher similarity** to human reasoning traces than those for **incorrect answers** across 5 of 6 models. For instance, GPT-5.1 demonstrates a substantial similarity gap of **+0.066** (**p < 0.001**, see the table below and **newly added Figure 23** in the revised paper). This confirms that while model reasoning generally differs from that of humans, correctness in SIV-Bench necessitates a convergence toward human-like social reasoning.
>
> | Model          | Similarity (Correct) | Similarity (Incorrect) | Gap (Δ) | Significance (p-value)      |
> |---|---|--|---|----|
> | GPT-5.1        | 0.572                | 0.506                    | +0.066   | **< 0.001 ($\*\*\*$)**          |
> | Qwen2.5-VL-7B  | 0.535                | 0.474                    | +0.061   | **< 0.01 ($\*\*$)**            |
> | GPT-4o-mini    | 0.571                | 0.513                    | +0.058   | **< 0.01 ($\*\*$)**            |
> | Gemini-2.5-Pro | 0.541                | 0.498                    | +0.043   | **< 0.05 ($\*$)**             |
> | Gemini-2.5-Flash | 0.537              | 0.504                    | +0.033   | **< 0.05 ($\*$)**             |
> | Gemini-3-Pro   | 0.527                | 0.494                    | +0.033   | 0.067 ($n.s.$)               |
>
> We have added **Appendix E** in the revised paper to include all the above results and analysis about **SIV-Bench-Hard**.
>
> > 2. Quantitative analysis of the Relation Inference task
>
> To provide a more rigorous understanding of model limitations and complement the qualitative failure analysis in Section 3.3 with quantitative evidence, we perform a deep-dive analysis on the critical **Relation Inference (RI)** subtask. This analysis decomposes performance across our **14 foundational relation types**.
>
> **Per-Relation Breakdown**: We have added **Table 11 (Appendix D.2)** to the revised paper, which details the accuracy of MLLMs across all 14 relation types. This breakdown reveals models' performance variance with nuanced relations (e.g., `Caregiver-Recipient`) compared to visually distinct ones.
>
> **Confusion Matrix Analysis**: To quantify failure patterns, we compute an aggregated Confusion Matrix **(Figure 19)**. This analysis exposes specific failure tendencies rooted in social context, such as:
> - **Inter-Group Confusion between Authority and Equality**: Models frequently confuse hierarchical roles with egalitarian ones (e.g., misclassifying Boss-Employee as Colleagues).
> - **Intra-Group Confusion**: Models often recognize general affect but fail to distinguish specific roles within a category (e.g., confusing Caregiver with Parent in Communal Sharing).
>
> This quantitative analysis provides granular evidence regarding model capabilities on specific social dimensions and empirically verifies the failure modes discussed in the main text.

---

### Note · Authors · 2026-01-02

**Comment:**

We thank the reviewers for their time and constructive feedback.

**Withdrawal Confirmation:**

I have read and agree with the venue's withdrawal policy on behalf of myself and my co-authors.